# AN ENHANCED GROMOV-WASSERSTEIN BARYCENTER METHOD FOR GRAPH-BASED CLUSTERING

## ABSTRACT

Optimal Transport (OT) recently has gained remarkable success in machine learning. These methods based on the Gromov-Wasserstein (GW) distance have proven highly effective in capturing complex data topologies and underlying structures. More specifically, Gromov-Wasserstein Learning (GWL) has recently introduced a framework for graph partitioning by minimizing the GW distance. Various improved versions stemming from this framework have showcased state-of-the-art performance on clustering tasks. Building upon GW barycenter, we introduce a novel approach that significantly enhances other GW-based models flexibility by relaxing the target distribution (cluster size) in GWL and using a wide class of positive semi-definite matrices. We then develop an efficient algorithm to solve the resulting non-convex problem by utilizing regularization and the successive upper-bound minimization techniques. The proposed method exhibits the capacity to identify improved partition results within an enriched searching space, as validated by our developed theoretical framework and numerical experiments. Furthermore, we bridge the proposed model with the well-known clustering methods including Non-negative Matrix Factorization, Min-Cut, Max-Dicut and other GW-based models. This connection provides a new solution to these classical clustering problems from the perspective of OT. Real data experiments illustrate our method outperforms state-of-the-art graph partitioning methods on both directed and undirected graphs.

## 1 INTRODUCTION

The Gromov-Wasserstein (GW) distance (Sturm, 2006; Mémoli, 2011), as an extension of classical optimal transport distance (Monge, 1781; Kantorovich, 1965), is based on the concept of Gromov-Hausdorff distance (Edwards, 1975), which measures the distance between different metric spaces. It is particularly useful for analyzing and comparing complex structures such as graphs (Peyré & Cuturi, 2019), where the classical Euclidean distance does not apply. In the case of graphs, the Gromov-Wasserstein distance measures the dissimilarity/similarity by minimizing the cost of mapping node pairs from one graph to node pairs in another graph. The distance preserving method between pairs of data points has been widely used in manifold learning such as tSNE (Van der Maaten & Hinton, 2008) for dimensionality reduction and this characteristic theoretically underpins the applications of GW distance in the analysis of structural data, such as object matching (Mémoli, 2011; Mémoli & Sapiro, 2004), analysis in biological and social networks. Despite the computationally demanding nature of GW distance based approaches, cutting-edge progress in machine learning and optimization methods is paving the way to surmount these obstacles (Peyré et al., 2016; Xu et al., 2019b). Consequently, GW distance is emerging as an increasingly crucial instrument for scrutinizing intricate structures spanning various domains.

**GW for graph partitioning** In this paper, our primary focus is on investigating the utilization of the GW distance in graph partitioning (graph-based clustering). Firstly, we consider a disconnected graph $G_1$ consisting of $K$ isolated and self-connected super-nodes as an ideal partitioned graph. Subsequently, we leverage the GW distance to measure the dissimilarity between the data graph $G_0(D_0, p_0)$ and the disconnected graph $G_1(D_1, p_1)$, where $p_i$ is the node distribution and $D_i$ is the node distance matrix for $i = 0, 1$. The induced optimal transport plan yields a soft mapping between nodes, facilitating the establishment of a $K-$way partition for $G_0$. Building upon this concept,

Xu et al. (2019a) introduced the Scalable Gromov-Wasserstein Learning (GWL) framework for graph partitioning, primarily relying on the graph's adjacency matrix. However, a pertinent question remains open: What is the optimal choice for matrix $D_0$ in this context? Notably, Chowdhury & Needham (2021) has shown that substituting the adjacency matrix with the heat kernel (SpecGWL) can lead to improved numerical results, all while preserving theoretical guarantees. However, both GWL and SpecGWL necessitate the prior estimation of the target distribution (cluster size) $p_1$ for the partitioned graph $G_1$. This requirement presents a significant challenge, as obtaining an accurate estimate can be exceedingly difficult. A recent breakthrough introduced in the form of Semi-Relaxed Gromov-Wasserstein graph partitioning (srGW) (Vincent-Cuaz et al., 2022) addresses this challenge by introducing a relaxation of this variable, resulting in improved performance in the context of clustering problems.

Despite recent strides in the field, applications of GW in graph modeling still grapple with substantial limitations. Firstly, these methods are constrained by variables like $p_1$ and $D_1$, restricting their adaptability and broader applicability. Secondly, the complex optimization problems they entail, often characterized by non-convex quadratic programs, pose challenges to efficiency. Finally, despite the great numerical performance, the absence of formal convergence guarantees has left questions about their reliability unanswered.

**Contributions**    We enhance model flexibility by relaxing the constraints of $p_1$ and $D_1$, offering a GW-based clustering framework with increased degrees of freedom, akin to a special Gromov-Wasserstein barycenter problem. Moreover, all current GW-based models can be seen as degenerate versions of our approach. The resulting optimization problem can be efficiently addressed within the alternating direction descent framework, employing successive upper-bound minimization (SUM) techniques and the incorporation of two novel regularization terms. The proposed algorithm theoretically guarantees finite convergence towards a transport mapping. We also demonstrate the state-of-the-art results of our algorithm in graph partitioning, underscoring its practical utility. Additionally, we establish significant connections between our algorithm and established classical clustering methods, including Non-negative Matrix Factorization (NMF), Min-Cut in undirected graphs, and Max-DiCut in directed graphs, opening a gate for new solutions to these classical models.

## 2    GRAPH PARITIONING BASED ON GW

**Gromov-Wasserstein discrepancy**    A dataset equipped with a graph structure with $n$ vertices can be represented as $G(D, \mu)$, where $[D]_{ij} \in \mathbb{R}^{n \times n}$ is a metric matrix characterizing the relationship between vertices, and $\mu \in \triangle_n = \{\mu \in \mathbb{R}^n_+ | \sum_{i=1}^n \mu_i = 1\}$ is a discrete probability distribution that characterizes the importance of the vertices. Given two graphs $G(D, \mu)$ and $G'(D', \nu)$ with $(D, \mu) \in \mathbb{R}^{N_1 \times N_1} \times \triangle_{N_1}, (D', \nu) \in \mathbb{R}^{N_2 \times N_2} \times \triangle_{N_2}$, the GW distance is defined as:

$$GW(G, G') = \min_{\pi \in \Pi(\mu, \nu)} \mathcal{E}_{D, D'}(\pi), \tag{1}$$

where

$$\mathcal{E}_{D, D'}(\pi) \stackrel{\text{def.}}{=} \sum_{i, i'} \sum_{j, j'} \left( D_{ii'} - D'_{jj'} \right)^2 \pi_{ij} \pi_{i'j'},$$

$$\Pi(\mu, \nu) \stackrel{\text{def.}}{=} \{\pi : \pi \in \mathbb{R}^{N_1 \times N_2}_+, \pi \mathbf{1}_{N_2} = \mu, \pi^\top \mathbf{1}_{N_1} = \nu\}.$$

Here $\mathbf{1}_N$ is a $N \times 1$ all-ones vector. And the concept of GW can be generalized to a pseudo-metric by incorporating any square matrix $D$ (Chowdhury & Mémoli, 2019) . GW distance measures the "cost" of transforming one graph into another while preserving its inherent topological structure by considering pairwise distances between elements, and the induced transport plan $\pi^*$ provides a tangible representation of the relationships between their nodes, making it a powerful tool for graph analysis and graph matching.

**Gromov-Wasserstein Learning**    In recent times, the GWL framework, (Xu et al., 2019a) has proven to be a valuable approach for tackling graph partitioning challenges. In a more specific context, when dealing with a data graph $G(D, \mu)$ consisting of $N$ nodes and a disconnected graph $G'(D', \nu)$ with a $K$ nodes where $N >> K$, we can view each node in $G'$ as a representative of a

distinct cluster, and the transport plan induced by GW discrepancy naturally provides indications of the probabilities of each node in association with each cluster (soft clustering), and its maximum posteriori actually implies a $K-$way partition (hard clustering) of graph $G$.

The core of the GWL framework (Xu et al., 2019a) is the representation of the data graph $G$ using an adjacency matrix $D = Adj$ and the characterization of $\mu$ through a function of node degree (neighbor density). Subsequently, an estimated distribution $\hat{\nu}$ and estimated structure matrix $\hat{D}' = diag(\hat{\nu})$ is derived by aggregating the weighted and resampled node distributions of $\mu$. The primary objective of $K-$way graph partitioning on a data graph comprising $N$ nodes is to optimize the Gromov-Wasserstein discrepancy problem with a predefined $\hat{D}'$ and $\hat{\nu}$. This optimization problem is formally expressed as:

$$\pi = \arg \min_{\pi \in \Pi(\mu, \hat{\nu})} \mathcal{E}_{Adj, \hat{D}'}(\pi). \tag{2}$$

**Heat kernels**    Bai & Hancock (2004) demonstrated that heat kernels may provide a more effective representation of the global structure of graphs in comparison with the adjacency matrix. For instance, in some cases, two graphs may have different edge connectivities but similar heat kernel structures, indicating that they have similar global properties despite their local differences. In contrast, adjacency-based matchings is based on only local information and may miss such global structures. The heat kernel can be computed by

$$H = e^{-tL} = \Phi e^{-t\Lambda} \Phi^\top, \tag{3}$$

and the problem is formally expressed as:

$$\pi = \arg \min_{\pi \in \Pi(\mu, \hat{\nu})} \mathcal{E}_{H, \hat{D}'}(\pi). \tag{4}$$

where $L$ is the graph Laplacian, and $\Lambda, \Phi$ are its eigenvalues and corresponding eigenvectors matrix. This can be understood from the perspective of heat equation, whose evolution operator is exactly (3). The heat kernel provides insights into how information or energy propagates and diffuses throughout the graph. Moreover, when $t \to 0$, we have $D \approx I - tL$, the kernel's properties are dictated by the local connectivity structure or topology of the graph. On the other hand, when $t \to \infty$, larger eigenvalues decay rapidly, resulting in $D \approx e^{-t\lambda_s} \phi_s \phi_s^\top$, where $\lambda_s$ is the smallest nonzero eigenvalue and $\phi_s$ is the corresponding eigenvector, namely Fiedler vector (Fiedler, 1973). Consequently, the global structure of the graph governs the kernel's behavior over long periods of time. Indeed, replacing the adjacency matrix with the heat kernel in GWL framework (SpecGWL) has been shown to achieve better results in graph partitioning (Chowdhury & Needham, 2021).

**semi-relaxed Gromov-Wasserstein**    A significant constraint inherent to both GWL and SpecGWL lies in their reliance on certain presuppositions. Specifically, they often require the introduction of a predetermined partitioned structure $\hat{D}'$ and a pre-established partitioned distribution $\hat{\nu}$. In the absence of prior knowledge pertaining to the accurate clustering of data (which fundamentally constitutes our initial challenge), providing a precise a priori estimation becomes an intricate undertaking. Existing estimation methods may be suboptimal across diverse scenarios, and can not accurately capture the true underlying structures within the graph. So Vincent-Cuaz et al. (2022) proposed semi-relaxed Gromov-Wasserstein (srGW) by relaxing the mass constraint on $\nu$ of GW discrepancy, and set $D' = \mathbb{I}_K$, then the problem is formally expressed as:

$$\pi = \arg \min_{\pi \in \mathbb{R}_+^{N \times K}, \pi \mathbf{1}_K = \mu} \mathcal{E}_{D, \mathbb{I}_K}(\pi). \tag{5}$$

# 3    ENHANCED GRAPH PARTITIONING METHOD BASED ON GROMOV-WASSERSTEIN BARYCENTER

## 3.1    A NEW GRAPH PARTITIONING MODEL

**Motivations**    In the realm of GW-based methods, certain limitations have been encountered, ranging from inherent model constraints to the absence of theoretical guarantees regarding algorithm convergence. Additionally, during our empirical investigations, we observed several intriguing phenomena.

Typically, we would generate a soft clustering represented by $\pi$ and then proceed to convert it into hard clustering assignments $\Gamma$ by utilizing the coordinates of each row's maximum. what piqued our interest was a noteworthy revelation: the truncated hard clustering consistently exhibited smaller GW distances than the soft clustering, i.e., $\mathcal{E}_{D,D'}(\Gamma) < \mathcal{E}_{D,D'}(\pi)$. This phenomenon sparked our curiosity and led us to contemplate the pursuit of hard clustering directly, a direction more closely aligned with the fundamental objective of clustering problems (more motivations in Appendix A). Figure 1 illustrates the GW distance obtained using SpecGWL with a heat kernel matrix for varying values of the hyperparameter $t$ for Wikipedia dataset.

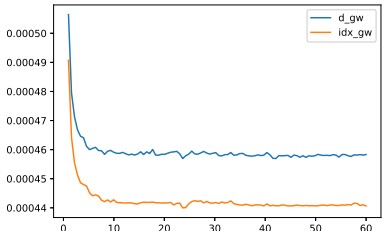

Figure 1: The GW distances obtained using SpecGWL with a heat kernel matrix for varying values of the hyperparameter $t$ for Wikipedia dataset. $d_{gw} = \mathcal{E}_{D,D'}(\pi)$ represents the distance induced by the soft clustering $\pi$. $idx_{gw} = \mathcal{E}_{D,D'}(\Gamma)$ signifies the GW distance induced after applying thresholding to $\pi$ for hard clustering $\Gamma$.

**Monge's type Gromov-Wasserstein barycenter** In the context of the $K-$way graph partitioning on a data graph $G(D, \mu)$ with N nodes, we naturally aspire to discover a **mapping** $\Gamma \in \mathbb{R}_+^{N \times K}$ from $G(D, \mu)$ to the ideal partitioned graph $G'(D', \nu)$, while minimizing the $GW(G, G')$. It's worth noting that for this mapping $\Gamma$, the $i$-th row contains solely one non-zero element, which equals $\mu_i$. If this element belongs to the $j$-th column, it signifies that the $i$-th node has been allocated to the $j$-th column. Consequently, the problem is reformulated as:

$$\min_{D' \in Diag_+} \min_{\Gamma \mathbf{1}_K = \mu, \|\Gamma\|_0 = N} \mathcal{E}_{D,D'}(\Gamma), \tag{6}$$

where $Diag_+$ represents the set of positive semidefinite diagonal matrix and $\|\cdot\|_0$ denotes the 0 norm, representing the number of non-zero elements, and induced optimal mapping $\Gamma$ can also be denoted as $MGW(G, G')$. It is worth noting that in this context, the entire structure $G'(D', \nu)$ is not pre-estimated; instead, it is treated as an optimization variable. Specifically, following common practice, the target structure $D'$ is often assumed to be a diagonal matrix (with non-negative diagonal elements, ensuring it is positive semi-definite), and the target distribution $\nu = \pi^T \mathbf{1}_N$ can be directly calculated. We can also reformulate this problem as a special case of Monge's type Gromov-Wasserstein barycenter problem:

$$G' = \arg \min_{G'(D', \nu), D' \in Diag_+} MGW(G, G'). \tag{7}$$

**Kantorovich's relaxation** Equation (6) involves the derivation of a hard clustering result by solving an intractable combinatorial optimization problem. Given its exceedingly high computational intricacy, it becomes imperative to initially embark on a process of relaxation to derive a continuous optimization problem. We can follow the approach used in OT and transform the Monge's type Gromov-Wasserstein problem into the continuous Kantorovich's type Gromov-Wasserstein problem, where we called Enhanced Gromov-Wasserstein Barycenter (EGWB):

$$D', \pi = \arg \min_{D' \in Diag_+, \pi \mathbf{1}_K = \mu} \mathcal{E}_{D,D'}(\pi). \tag{8}$$

We will only consider local minimizer of this problem at the extremal points of $\Pi(\mu, \cdot) = \{\pi \in \mathbb{R}_+^{N \times K}, \pi \mathbf{1}_K = \mu\}$, which precisely constitutes the feasible sets of the mapping $\Gamma$. Furthermore, in the discrete case, any distribution can be transformed into a uniform distribution through splitting

of mass, and we can prove that problem (8) achieves its global minimum precisely at the extremal points under some conditions (see Appendix A), therefore we will consider the case where $\mu$ follows a uniform distribution in the following sections.

## 3.2 OPTIMIZATION AND ALGORITHM

We can employ alternating minimization to solve (8), in details, starting with an initial guess $\pi(0)$, then we compute the sequence:

$$D^{'}(0), \pi(1), D^{'}(1), \pi(2), D^{'}(2), \cdots, D^{'}(t), \pi(t+1), \cdots \tag{9}$$

**Updating $D^{'}$ with fixed $\pi$**    The subproblem can be reformulated as:

$$D^{'}(t) = \arg\min_{D^{'}} \sum_{j,j'} (D^{'}_{jj'}{}^2 \sum_{i,i'} \pi_{ij}\pi_{i'j'} - 2D^{'}_{jj'} \sum_{i,i'} \pi_{ij}D_{ii'}\pi_{i'j'}),$$

which can be decoupled into independent subproblems with respect to $D^{'}_{jj'}$, and each problem is a quadratic convex function, we can efficiently obtain the minimum by

$$D^{'}_{jj}(t) = \begin{cases} \frac{\sum_{i,i'} \pi_{ij}D_{ii'}\pi_{i'j}}{\sum_{i,i'} \pi_{ij}\pi_{i'j}} = \frac{\pi_{:,j}^{\top}D\pi_{:,j}}{\nu_j^2}, & \text{for } \nu_j > 0 \\ -c, & \text{otherwise} \end{cases} \tag{10}$$

where $\pi_{:,j}$ is the $j-$th column vector of $\pi$, $\nu_j$ is the probability of $j$ cluster and $c$ is a large constant.

**Updating $\pi$ with fixed $D^{'}$**    This subproblem usually is a non-convex optimization problem (see Appendix A) with respect to $\pi \in \Pi(\frac{1_N}{N}, \cdot)$, so discovering a global minimum remains an unattainable pursuit. Note that the original problem (6) concerns hard clustering, thus our aspiration lies in identifying local minimizer on the extremal points, which is given by $\frac{1}{N}\text{Ind}^{N \times K}$ with $\text{Ind}^{N \times K}$ representing the set of all rectangular permutation matrices (Cao et al., 2022). In summary, we want to find

$$\pi(t+1) = \arg\min_{\pi \in \frac{1}{N}\text{Ind}^{N \times K}} \mathcal{E}_{D,D'}(\pi), \tag{11}$$

where $D^{'} = D^{'}(t)$. For such a class of problems, methods like Conditional Gradient (CG) algorithm (Jaggi, 2013) can be employed for solving. Benefitting from the inherent peculiarity of the problem structure, it is guaranteed to converge to extremal points eventually (Lacoste-Julien, 2016). However, numerically, it falls short in comparison to methods that incorporate entropy regularization (Vincent-Cuaz et al., 2022). On the other hand, we can observe the fact that adding a constant $-\lambda\frac{1}{N} = -\lambda\langle\pi, \pi\rangle$ to (18) does not affect the optimizer for any $\lambda$ and $\pi \in \frac{1}{N}\text{Ind}^{N \times K}$, where $\langle\cdot, \cdot\rangle$ is Frobenius inner product. Thus, we can solve the corresponding continuous Kantorovich's type problem:

$$\pi(t+1) = \arg\min_{\pi \in \Pi(\frac{1_N}{N}, \cdot)} f(\pi) = \arg\min_{\pi \in \Pi(\frac{1_N}{N}, \cdot)} (\mathcal{E}_{D,D'}(\pi) - \lambda\langle\pi, \pi\rangle), \tag{12}$$

with some certain $\lambda \geq 0$. Additionally, inspired by the technique of entropy regularization, we can solve (12) by generating a sequence $\{\pi^{\tau, \varepsilon}\}$ that successively minimizing the approximate upper bound function of $f$:

$$\begin{aligned} \pi^{\tau+1, \varepsilon} &= \arg\min_{\pi \in \Pi(\frac{1_N}{N}, \cdot)} u(\pi, \pi^{\tau, \varepsilon}) \\ &= \arg\min_{\pi \in \Pi(\frac{1_N}{N}, \cdot)} f(\pi^{\tau, \varepsilon}) + \langle\nabla f(\pi^{\tau, \varepsilon}), \pi - \pi^{\tau, \varepsilon}\rangle + \varepsilon D_{\text{KL}}(\pi\|\pi^{\tau, \varepsilon}). \end{aligned}$$

This process can be regarded as proximal gradient method, and be reformulated as an Eulerian discrete Wasserstein Barycenter problem (Peyré & Cuturi, 2019) with an entropy regularizer:

$$\pi^{\tau+1, \varepsilon} = \arg\min_{\pi \in \Pi(\frac{1_N}{N}, \cdot)} \underbrace{\langle\nabla f(\pi^{\tau, \varepsilon}) - \varepsilon ln(\pi^{\tau, \varepsilon}), \pi\rangle}_{C'} + \varepsilon\langle\pi, ln(\pi)\rangle. \tag{13}$$

By the first-order optimality condition and constraints, it can be directly obtained using the one-step Sinkhorn projection (detailed in Appendix A), as shown in Algorithm 1. It's pertinent to note that at each iteration, the computation of the gradient entails a complexity of $O(N^2K + K^2N)$, while a single Sinkhorn projection carries a complexity of $O(NK)$.

---

**Algorithm 1:** EGWB-$\pi$

---

**Input:** Similarity matrix in data space: D, previous $D^{'},\varepsilon,\lambda$
**Output:** $\pi_{new}$.

1 Initialize $\pi^0$;
2 **while** not converged **do**
3 $\quad$ Evaluate Cost matrix $C^{'} = \nabla f(\pi^{\tau,\varepsilon}) - \varepsilon ln(\pi^{\tau,\varepsilon})$;
4 $\quad$ Evaluate Kernel $K = exp(-\frac{C^{'}}{\varepsilon})$;
5 $\quad$ Calculate transport plan $\pi^{\tau+1,\varepsilon} = \frac{K_{i,j}}{N\sum_{j=1}^{k} K_{ij}}$;
6 **end**
7 $\pi_{new} = \pi^{\tau,\varepsilon}$.

---

By introducing a strongly concave regularization term that $-\langle\pi,\pi\rangle$ renders the objective function "more" concave, and subsequently employing a convex regularization term $\varepsilon D_{\mathrm{KL}}(\pi\|\pi^{\tau,\varepsilon})$ to provide a lower bound for the coefficient $\lambda$ associated with the concave regularization term, our algorithm 1 eventually converges to the extremal points.

**Theorem 1 (Convergence for Algorithm 1)** *If $\lambda \geq N \times max(D^{'})^2 - \frac{\varepsilon N}{4}$, every limiting point generated by the sequence $\{\pi^{\tau,\varepsilon}\}$ is a stationary point of Problem* (12).

It is straightforward to observe the monotone decrease of the objective function during the alternating minimization process. As the objective function has a lower bound, we can infer the convergence.

**Theorem 2 (Monotone decrease of the objective function)** *The sequences $\{D^{'}(t)\}$ and $\{\pi(t)\}$ in process* (9) *satisfies:*

$$\mathcal{E}_{D,D^{'}(t)}(\pi(t+1)) \leq \mathcal{E}_{D,D^{'}(t)}(\pi(t)) \leq \mathcal{E}_{D,D^{'}(t-1)}(\pi(t)) \quad \forall t.$$

**Theorem 3 (Convergence for process** (9)**)** *If $\lambda > N \times max(D^{'})^2 - \frac{\varepsilon N}{4}$, then the cumulative error $\sum_{\tau}\|\pi^{\tau,\varepsilon} - \pi^{\tau+1,\varepsilon}\|_F^2$ is bounded by $|\mathcal{E}_{D,D^{'}(t)}(\pi(t+1)) - \mathcal{E}_{D,D^{'}(t)}(\pi(t))|$ in Algorithm 1. Then for sufficiently large t, $\{\pi^{\tau,0}\}$ is a constant sequence, inferring the convergence of $\pi(t)$.*

In general, we can not prove the global convergence for $\varepsilon > 0$. However, in our numerical experiments we found that a small enough $\varepsilon$ always leads to the convergence of $\{\pi(t)\}$.

**Highlights** Our proposed algorithms 1 and process (9) combine the strengths of conditional gradient method and entropy regularization. By introducing a concave regularization term, we maintain the GW distance value at the extremal points while increasing its value in the interior of the joint probability space $\Pi(\mu,\cdot)$. This design ensures that our algorithm converges to extremal points, effectively forming a hard clustering result. Furthermore, we incorporate a KL divergence regularization term to prevent premature convergence to a poor local minima, thus have better numerical performance than CG. Additionally, convergence properties of our new algorithms are theoretically guaranteed.

### 3.3 CONNECTIONS WITH OTHER METHODS

**GW-based methods** In the context of our EGWB framework, when either $D^{'}$, $\nu$, or both are fixed, our method simplifies to other GW-based approaches like GWL, SpecGWL, and srGW (detailed in Appendix B).

- By fixing $\nu = \hat{\nu}$ and specifying $D^{'} = diag(\hat{\nu})$, with $\lambda$ assigned the constant value of 0, the proposed $\pi$ subproblem 12 can be regarded as equivalent to either **GWL** or **SpecGWL**, contingent on our choice between the adjacency matrix or the heat kernel matrix for $D$.

- By fixing $D^{'} = \mathbb{I}_K$, with $\lambda$ assigned the constant value of 0, the proposed $\pi$ subproblem can be regarded as equivalent to **srGW**. Particularly, the objective function is equivalent to a

vanilla Min-Cut function plus an exclusive lasso (Zhou et al., 2010) regularization term with a coefficient of $\frac{1}{2}$. This results in each cluster containing an equal number of data points. As we further optimize the regularization term coefficient, it evolves into a **balanced Min-Cut** problem (Chen et al., 2017) .

**Other benchmark methods**   Our model can also be reformulated into other clustering models, and we introduce a novel relaxation technique from OT viewpoint to optimize it within the probability space for $\pi$ (detailed in Appendix B).

- The objective function of EGWB is equivalent to **enhanced balanced Min-Cut** (Chen et al., 2020). In particular, if we substitute the optimal $D'$ obtained in the previous step (10) into the $\pi$ subproblem, we can get

$$\pi^* = \arg\min_{\pi} \sum_{l=1}^{K} -\frac{(\pi_{:,l}^\top D \pi_{:,l})^2}{|V_l|^2}, \qquad (14)$$

  which has a similar objective as the classical **normalized cut** problem (Shi & Malik, 2000). Our proposed EGWB involves maximizing the same within-cluster similarities, but utilize different normalization terms, which can also enhance the model's robustness when dealing with isolated nodes.

- After a straightforward derivation, the objective function (6) can also be reformulated as **Weighted symmetric NMF** form (Ding et al., 2005):

$$D', T = \arg\min_{D', T} \|D - TD'T^\top\|_F^2.$$

- When considering the partition of a directed graph, we can define the flow ratio (Laenen & Sun, 2020) by setting $D'$ as an upper triangular matrix with only the sub-diagonal elements being non-zero, and the objective function of EGWB is equivalent to a **Max-Dicut** problem:

$$\pi^* = \arg\max_{\pi} \sum_{i=1}^{K} \frac{\mathrm{cut}\,(V_i, V_{i+1})^2}{|V_i||V_{i+1}|}. \qquad (15)$$

## 4   EXPERIMENTS

We perform several numerical experiments using both synthetic and real-world datasets to demonstrate the superior performance of our algorithm.

### 4.1   SYNTHETIC DATA

We first construct four blobs, each containing 100, 200, 300, and 400 points, with varying standard deviations (0.55, 1.0, 1.5, and 2.0), illustrated in Figure 2a. We then apply the k-nearest neighbors (k-NN) method with $k = 200$ to construct the Gaussian kernel adjacency matrix for GWL and use $t = 100$ to construct the heat kernel matrix for SpecGWL. We consider two different prior cluster size: estimated values $\hat{\nu} = (0.25, 0.25, 0.25, 0.25)$ from Xu et al. (2019a) and the true values $\nu_{true} = (0.1, 0.2, 0.3, 0.4)$. The results are displayed in Figure 2. We see that SpecGWL outperforms GWL by utilizing the heat kernel instead of the adjacency matrix. On the other hand, the choice of prior estimates $\hat{\nu}$ significantly impacts clustering results for both GWL and SpecGWL, with better estimates yielding improved outcomes. Notably, even with the true values $\nu_{true}$, neither GWL nor SpecGWL achieve correct clustering results, while EGWB provides accurate results without prior information (detailed in Appendix B).

### 4.2   REAL DATA

We proceed to assess the efficacy of EGWB in comparison to other GW-based techniques such as GWL, SpecGWL, srGW, as well as several baseline methods including Infomap (Rosvall & Bergstrom, 2008), FluidC (Parés et al., 2018), and Newman Fast Algorithm (Newman, 2006).

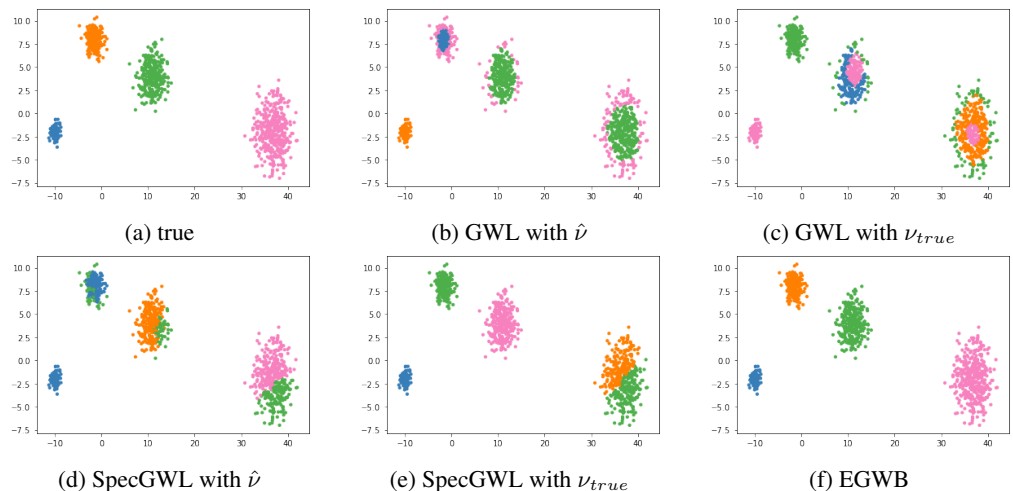

Figure 2: Clustering results. From left to right, 4 blobs with standard deviation 0.55, 1.0, 1.5, and 2.0, there are 100, 200, 300, and 400 points respectively. Different colors represent different clusters.

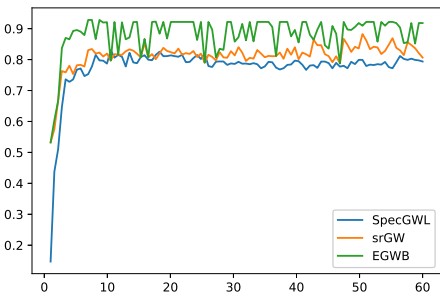

Figure 3: Comparison of AMI results between SpecGWL, srGW, and EGWB indicates that EGWB performs the best. This aligns with our theoretical analysis, where SpecGWL and srGW are degenerate models of SpecGWL.

**Datasets** Here, we utilized four classic real-life datasets as used in Chowdhury & Needham (2021). The first dataset is **Wikipedia** hyperlink network (Leskovec & Krevl, 2014), where the digraph has 1998 nodes and 2700 edges, with 15 categories. The second dataset is derived from a network of **Amazon** products (Leskovec & Krevl, 2014) with 1501 nodes and 4626 edges, with 12 categories. The third dataset originates from a **European** research institution. It comprises 1005 nodes, representing the 42 departments and their members, and 25571 edges. The final is **Village** dataset representing a social network which captures the 8423 interactions (edges) among 1991 residents of 12 villages in rural India (Banerjee et al., 2013). We also generate a noisy version of each graph by adding up to 10% randomly generated additional edges.

**Experimental settings** We constructed the heat kernel matrix using the normalized graph Laplacian for undirected graphs and Chung's normalized Laplacian for directed graphs. Furthermore, all hyperparameters, such as the heat kernel time $t$ and entropy regularization coefficient $\epsilon$, can be tuned using maximum modularity. Additionally, with respect to our newly introduced parameter $\lambda$, when $\lambda$ becomes exceedingly large, the concave regularization term dominates the objective function, causing $f(\pi)$ to become a concave function. Consequently, Algorithm 1 rapidly converges to extremal points but highly dependent on initial conditions. To mitigate this issue, we employed a widely-used mathematical technique known as a **continuation scheme**. In this scheme, we gradually increased the value of $\lambda$ from a small value to a larger one. Intuitively, in the initial stages of the algorithm when $\lambda$ is small (close to 0), it resembles classical entropy regularization methods, searching for

Table 1: Comparison of $AMI$ across a variety of datasets

|  | Wikipedia | | EU-email | | Amazon | | Village | |
|---|---|---|---|---|---|---|---|---|
|  | noisy | raw | noisy | raw | noisy | raw | noisy | raw |
| EGWB (ours) | 0.526 | 0.570 | 0.511 | 0.565 | 0.587 | 0.780 | 0.833 | 0.923 |
| GWL | 0.332 | 0.438 | 0.357 | 0.411 | 0.346 | 0.414 | 0.509 | 0.688 |
| SpecGWL | 0.473 | 0.510 | 0.443 | 0.493 | 0.441 | 0.605 | 0.747 | 0.822 |
| srGW | 0.505 | 0.572 | 0.466 | 0.534 | 0.532 | 0.700 | 0.711 | 0.865 |
| Fluid | 0.347 | NA | 0.450 | NA | 0.182 | NA | 0.486 | NA |
| Newman | 0.341 | 0.382 | 0.231 | 0.312 | 0.668 | 0.772 | 0.721 | 0.880 |
| InfoMap | 0.329 | 0.377 | 0.350 | 0.447 | 0.518 | 0.942 | 0.162 | 0.882 |

more directions within the feasible set. As $\lambda$ increases, our algorithm eventually converges to an extremal point. The detailed sensitivity analyses for all parameters are provided in Appendix B.

**Results and discussion** We employ five different metrics, Their performance exhibits striking similarities. In this context, we employ Adjusted Mutual Information (AMI) as a representative metric to assess the quality of our clustering results, as illustrated in Table 1 and Table 2. Additional results are provided in Appendix B. Specifically, Figure 3 illustrates the results of SpecGWL, srGW, and EGWB concerning different heat kernel times $t$. It can be observed that EGWB consistently outperforms other GW-based methods, thereby confirming the theoretical analysis that EGWB can be reduced to other GW-based approaches. It is worth noting that on the Wikipedia dataset, srGW and EGWB demonstrate similar performance. This can be attributed to the fact that Wikipedia's data is equally partitioned, and the exclusive lasso term in srGW facilitates rapid convergence to a perfect clustering result. In comparison to other baseline methods: FluidC is notably limited as it is not applicable to disconnected graphs; EGWB's performance is on par with Infomap on networks with high density such as the Amazon and Village datasets, and even surpasses Infomap on both sparse networks like the Wikipedia dataset and extremely dense networks like the Eu-email dataset; unlike the Newman Fast Algorithm which is exclusively suitable for unweighted graphs, EGWB can handle weighted graphs. However, for a fair comparison, we present results only for unweighted graphs, where Newman and EGWB exhibit similar performance. Furthermore, we want to stress that all GW-based methods demonstrate greater robustness compared to the other methods, as their results remain largely unchanged even after introducing noise.

Table 2: Comparison of $AMI$ across a variety of digraph datasets

|  | Wikipedia | | EU-email | |
|---|---|---|---|---|
|  | noisy | raw | noisy | raw |
| EGWB (ours) | 0.486 | 0.516 | 0.500 | 0.585 |
| GWL | 0.165 | 0.201 | 0.341 | 0.422 |
| SpecGWL | 0.336 | 0.399 | 0.385 | 0.452 |
| srGW | 0.494 | 0.525 | 0. 537 | 0.563 |
| InfoMap | 0.356 | 0.376 | 0.455 | 0.584 |

## 5 CONCLUSION AND FUTURE WORK

In conclusion, we have introduced a novel graph partitioning method based on the Monge's type Gromov-Wasserstein barycenter. This approach enhances the flexibility of existing GW-based models by relaxing both the target structure and distribution constraints. We have also devised an efficient algorithm to tackle the ensuing complex optimization problem, which incorporates two regularization terms and utilizes the SUM technique, with theoretical guarantees ensuring its convergence. Furthermore, we have demonstrated that under certain parameter settings, the proposed model reduces to existing GW-based methods. This has been confirmed by our numerical experiments conducted on synthetic and real datasets, consistently showcasing EGWB as the superior choice among GW-based methods. Moreover, our work has established a meaningful bridge between our proposed model and classical clustering techniques such as Non-negative Matrix Factorization,

Min-Cut, and Max-Dicut. This connection offers fresh perspectives on addressing classic clustering problems from an Optimal Transport standpoint. In the future, we plan to explore the incorporation of both edge and node information in our graph-based clustering models, as inspired by recent studies such as FusedGW (Vayer et al., 2020) and other relevant works (Neyshabur et al., 2013; Vijayan et al., 2015; Sun et al., 2015). Furthermore, we are eager to apply our model and algorithm to a broader range of intriguing tasks, such as the clustering of graph datasets and graph completion.

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

# A    APPENDIX

## A.1    MATHEMATICAL PRELIMINARIES AND DISCUSSIONS

**Non-convexity of GW discrepancy**    For any fixed graphs $G(D, \mu)$ and $G'(D', \nu)$, we can reformulate the GW-discrepancy as following:

$$
\begin{aligned}
GW_2(G, G') &= \min_{\pi \in \Pi(\mu, \nu)} \sum_{i,i'} \sum_{j,j'} \left( D_{i,i'} - D'_{j,j'} \right)^2 \pi_{ij} \pi_{i'j'} \\
&= \min_{\pi \in \Pi(\mu, \nu)} \sum_{i,i'} D^2_{i,i'} \sum_j \pi_{ij} \sum_{j'} \pi_{i'j'} + \sum_{j,j'} (D'_{j,j'})^2 \sum_i \pi_{ij} \sum_{i'} \pi_{i'j'} \\
&\qquad\qquad -2 \sum_{i,i'} \sum_{j,j'} \pi_{ij} D_{i,i'} \pi_{i'j'} D'_{j,j'} \\
&= \min_{\pi \in \Pi(\mu, \nu)} \sum_{i,i'} D^2_{i,i'} \mu_i \mu_{i'} + \sum_{j,j'} (D'_{j,j'})^2 \nu_j \nu_{j'} \\
&\qquad\qquad -2 \sum_{j,i,i',j'} \pi^T_{ji} D_{i,i'} \pi_{i'j'} D'_{j',j} \\
&\iff \min_{\pi \in \Pi(\mu, \nu)} - Tr(\pi^T D \pi D').
\end{aligned}
$$

If $D$ and $D'$ are posite semidefinite, then we can find $U$ and $V$ such that $D = U^T U$, $D' = V^T V$, then $-Tr(\pi^T D \pi D') = -\|U \pi V^T\|^2_F$ is concave, the minimum can be attained at the extremal point of $\Pi(\mu, \nu)$. In our EGWB model, the $\nu$ is not fixed and its objective function can be reformulated as:

$$
\begin{aligned}
L(\pi) &= \min_{\pi \in \Pi(\mu, \cdot)} \sum_{i,i'} \sum_{j,j'} (D'_{j,j'})^2 \pi_{ij} \pi_{i'j'} - 2 \sum_{i,i'} \sum_{j,j'} \pi^T_{ji} D_{i,i'} \pi_{i'j'} D'_{j,j'} \\
&= \min_{\pi \in \Pi(\mu, \cdot)} Tr(\pi^T \mathbf{1} \mathbf{1}^T \pi \left( D' \odot D' \right)) - 2 Tr(\pi^T D \pi D'),
\end{aligned}
$$

and we have

$$
\begin{aligned}
\frac{dL}{d\pi} &= \frac{dTr(\pi^T \mathbf{1} \mathbf{1}^T \pi \left( D' \odot D' \right)) - 2 Tr(\pi^T D \pi D')}{d\pi} \\
&= 2 \mathbf{1} \mathbf{1}^T \pi \left( D' \odot D' \right) - 4 D \pi D' \\
\frac{dL^2}{d^2\pi} &= 2 \left( D' \odot D' \right) \bigotimes \left( \mathbf{1} \mathbf{1}^T \right) - 4 D' \bigotimes D,
\end{aligned}
$$

where the Hessian matrix is not positive semidefinite in general, thus making the $\pi$ subproblem non-convex.

**Assumption of uniform distribution.**    For a discrete probability distribution $\sum_{i=1}^N \mu_i \delta_{x_i}$, we can always decompose it into the form of uniform distribution. In fact, since the discrete probability distribution is the sum of some Dirac measures, we can always use the split mass method to find the smallest unit measure , for example, $\frac{1}{3} \delta_{-1} + \frac{2}{3} \delta_1$ can be divided into $\frac{1}{3} \delta_{-1} + \frac{1}{3} \delta_1 + \frac{1}{3} \delta_1$. We will show this split will keep the GW distance unchanged.

In fact, assuming that the index set of data points is represented as $I = I_1 \bigcup I_2$, where $I_1 = \{i_0\}$ denotes the index of point $x_{i_0}$ that require a mass split. We can split the point $x_{i_0}$ into a set of identical points $\{x_i\}_{i \in I'_1}$ with equal weights $\mu_i (i \in I'_1)$. They satisfy $\sum_{i \in I'_1} \mu_i = \mu_{i_0}$. Then for each transport plan $\pi$, the corresponding row $\pi_{i_0 \cdot}$ can be divided into $\sum_{i \in I_1'} \widetilde{\pi}_{i \cdot}$. Consequently, after the split, we obtain a new index set $I' = I'_1 \bigcup I_2$. Moreover, for the new point clouds, we define

$$
\widetilde{D}_{ii'} = \begin{cases} D_{i_0 i_0} & \text{for } i, i' \in I'_1, \\ D_{i_0 i'} & \text{for } i \in I_1, \ i' \in I_2, \\ D_{ii'} & \text{for } i \in I_2. \end{cases}
$$

then for each $\widetilde{\pi}$, we have

$$
\begin{aligned}
\mathcal{E}_{\widetilde{D},D'}(\widetilde{\pi}) &= \sum_{i\in I_1', i'\in I_1'}\sum_{j,j'}\left(\widetilde{D}_{ii'}-D'_{jj'}\right)^2\widetilde{\pi}_{ij}\widetilde{\pi}_{i'j'} + 2\sum_{i\in I_1', i'\in I_2}\sum_{j,j'}\left(\widetilde{D}_{ii'}-D'_{jj'}\right)^2\widetilde{\pi}_{ij}\widetilde{\pi}_{i'j'}\\
&\quad + \sum_{i\in I_2, i'\in I_2}\sum_{j,j'}\left(\widetilde{D}_{ii'}-D'_{jj'}\right)^2\widetilde{\pi}_{ij}\widetilde{\pi}_{i'j'}\\
&= \sum_{j,j'}\left(D_{i_0 i_0}-D'_{jj'}\right)^2\sum_{i\in I_1'}\widetilde{\pi}_{ij}\sum_{i'\in I_1'}\widetilde{\pi}_{i'j'} + 2\sum_{i'\in I_2}\sum_{j,j'}\left(D_{i_0 i'}-D'_{jj'}\right)^2\sum_{i\in I_1'}\widetilde{\pi}_{ij}\pi_{i'j'}\\
&\quad + \sum_{i\in I_2, i'\in I_2}\sum_{j,j'}\left(D_{ii'}-D'_{jj'}\right)^2\pi_{ij}\pi_{i'j'}\\
&= \sum_{i\in I_1, i'\in I_1}\sum_{j,j'}\left(D_{ii'}-D'_{jj'}\right)^2\pi_{ij}\pi_{i'j'} + 2\sum_{i\in I_1, i'\in I_2}\sum_{j,j'}\left(D_{ii'}-D'_{jj'}\right)^2\pi_{ij}\pi_{i'j'}\\
&\quad + \sum_{i\in I_2, i'\in I_2}\sum_{j,j'}\left(D_{ii'}-D'_{jj'}\right)^2\pi_{ij}\pi_{i'j'}\\
&= \mathcal{E}_{D,D'}(\pi)
\end{aligned}
$$

Therefore, without loss of generality, we always assume uniform distribution $\mu = \frac{\mathbf{1}_{N*1}}{N}$.

**One common condition for extremal points** When we consider the case where $\mu$ follows a uniform distribution and assume $D' \in \mathbb{S}_+^K$ is a positive semidefinite matrix, then the objective function is

$$
D', \pi = \arg\min_{D'\in Diag, \pi\in\Pi(\frac{\mathbf{1}_N}{N}, \cdot)} \mathcal{E}_{D,D'}(\pi), \tag{16}
$$

where $\Pi(\frac{\mathbf{1}_N}{N}, \cdot)$ represents one side constraint $\pi\mathbf{1}_K = \frac{\mathbf{1}_N}{N}$. Noted that the $\pi$ subproblem is a non-convex optimization problem with respect to $\pi\in\Pi(\frac{\mathbf{1}_N}{N}, \cdot)$, but we can reformulated it as

$$
\min_{\pi\in\Pi(\frac{\mathbf{1}_N}{N}, \cdot)} \mathcal{E}_{D,D'}(\pi) = \min_{\nu}\mathcal{E}_{D,D'}(\pi_\nu), \tag{17}
$$

where $\pi_\nu$ is the minimizer of $\mathcal{E}_{D,D'}(\pi)$ over $\Pi(\frac{\mathbf{1}_N}{N}, \nu)$ for any fixed $\nu$. Since $D$ and $D'$ are positive semidefinite, then $\exists U, V$ such that $D = U^\top U$, $D' = V^\top V$, then we have

$$
\pi_\nu = \arg\min_{\pi\in\Pi(\frac{\mathbf{1}_N}{N}, \nu)} -\|U\pi V^\top\|_F^2,
$$

where $-\|U\pi V^\top\|_F^2$ is a concave function, and the minimum $\pi_\nu^*$ can be obtained at an extremal point of $\Pi(\frac{\mathbf{1}_N}{N}, \nu)$. In the case $N\nu \in \mathbf{N}^K$, all extremal points form a set denoted by $A_\nu$, where $A_\nu$ contains matrices having exactly one $\frac{1}{N}$ in each row and a column sum of $\nu$. Then (17) is equivalent to minimizing $\mathcal{E}_{D,D'}(\pi)$ over $\bigcup_\nu A_\nu$, which is given by $\frac{1}{N}\mathrm{Ind}^{N\times K}$ with $\mathrm{Ind}^{N\times K}$ being the set of all rectangular permutation matrices Cao et al. (2022). In summary,

$$
\pi = \arg\min_{\pi\in\frac{1}{N}\mathrm{Ind}^{N\times K}} \mathcal{E}_{D,D'}(\pi). \tag{18}
$$

**Alternative condition for extremal points** Here we proposed another condition

$$
\max D' \le 2\min_i D_{ii} \tag{19}
$$

to replace the condition $N\nu \in \mathbf{N}^K$, under which $\mathcal{E}_{D,D'}(\pi)$ can also attain its minimum at the extremal points of $\Pi(\mu, \cdot)$. Let's assume the contrary, which means there exists a minimizing transport plan $\pi$, which is not an extremal point of $\Pi(\mu, \cdot)$. Specifically, there exists $i_0$ such that

$$
\pi_{i_0\cdot} = \mu_{i_0}[\theta_1 \quad \theta_2 \quad \cdots \quad \theta_K],
$$

where $\sum_{s=1}^{K} \theta_s = 1$ and $0 \leq \theta_s < 1$. Furthermore, we define $I_1 = \{i_0\}$ and $I_2 = I \setminus I_1$, where $I = \{1, 2, \cdots, N\}$ is the full index set. Then, we can define $\{\hat{\pi}(s)\}_{s=1}^{K}$ such that

$$\hat{\pi}(s)_{ij} = \begin{cases} \mu_{i_0} & \text{for } i \in I_1, j = s, \\ 0 & \text{for } i \in I_1, j \neq s, \\ \pi_{ij} & \text{for } i \in I_2. \end{cases}$$

Hence, the $i_0$-th row of $\hat{\pi}(s)$ corresponds to an extremal point, and $\pi$ can be expressed as a linear combination of $\hat{\pi}(s)$, specifically, $\pi = \sum_{s=1}^{K} \theta_s \hat{\pi}(s)$. Next, we aim to show that $\mathcal{E}_{D,D'}(\pi) \geq \sum_{s=1}^{K} \theta_s \mathcal{E}_{D,D'}(\hat{\pi}(s))$. If this condition is satisfied, then there exists an $s_0$ such that $\mathcal{E}_{D,D'}(\hat{\pi}(s_0)) \leq \mathcal{E}_{D,D'}(\pi)$. We can continue this procedure for other rows of $\pi$ which do not locate at the extremal points. Eventually we can obtain a transport plan $\tilde{\pi}$ locating at an extremal point of $\Pi(\mu, \cdot)$ with $\mathcal{E}_{D,D'}(\tilde{\pi}) \leq \mathcal{E}_{D,D'}(\pi)$, which leads to the contradiction. To prove $\mathcal{E}_{D,D'}(\pi) \geq \sum_{s=1}^{K} \theta_s \mathcal{E}_{D,D'}(\hat{\pi}(s))$, it is straightforward to find that,

$$\mathcal{E}_{D,D'}(\pi) - \sum_{s=1}^{K} \theta_s \mathcal{E}_{D,D'}(\hat{\pi}(s))$$

$$= \underbrace{\sum_{i \in I_1, i' \in I_1} \sum_{j,j'} \left(D_{ii'} - D'_{jj'}\right)^2 \pi_{ij} \pi_{i'j'} - \sum_{s=1}^{K} \theta_s \sum_{i \in I_1, i' \in I_1} \sum_{j,j'} \left(D_{ii'} - D'_{jj'}\right)^2 \hat{\pi}(s)_{ij} \hat{\pi}(s)_{i'j'}}_{I}$$

$$+ \underbrace{2 \sum_{i \in I_1, i' \in I_2} \sum_{j,j'} \left(D_{ii'} - D'_{jj'}\right)^2 \pi_{ij} \pi_{i'j'} - \sum_{s=1}^{K} 2\theta_s \sum_{i \in I_1, i' \in I_2} \sum_{j,j'} \left(D_{ii'} - D'_{jj'}\right)^2 \hat{\pi}(s)_{ij} \hat{\pi}(s)_{i'j'}}_{II}$$

$$+ \underbrace{\sum_{i \in I_2, i' \in I_2} \sum_{j,j'} \left(D_{ii'} - D'_{jj'}\right)^2 \pi_{ij} \pi_{i'j'} - \sum_{s=1}^{K} \theta_s \sum_{i \in I_2, i' \in I_2} \sum_{j,j'} \left(D_{ii'} - D'_{jj'}\right)^2 \hat{\pi}(s)_{ij} \hat{\pi}(s)_{i'j'}}_{III},$$

where

$$I = \sum_{j,j'} \left(D_{i_0 i_0} - D'_{jj'}\right)^2 \pi_{i_0 j} \pi_{i_0 j'} - \sum_{s=1}^{K} \theta_s \sum_{j,j'} \left(D_{i_0 i_0} - D'_{jj'}\right)^2 \hat{\pi}(s)_{i_0 j} \hat{\pi}(s)_{i_0 j'}$$

$$= (D_{i_0 i_0})^2 \sum_{j,j'} \pi_{i_0 j} \pi_{i_0 j'} + \sum_{j,j'} D'_{jj'} \left(D'_{jj'} - 2D_{i_0 i_0}\right) \pi_{i_0 j} \pi_{i_0 j'}$$

$$- \sum_{s=1}^{K} \theta_s (D_{i_0 i_0})^2 \sum_{j,j'} \hat{\pi}(s)_{i_0 j} \hat{\pi}(s)_{i_0 j'} - \sum_{s=1}^{K} \theta_s \sum_{j,j'} D'_{jj'} \left(D'_{jj'} - 2D_{i_0 i_0}\right) \hat{\pi}(s)_{i_0 j} \hat{\pi}(s)_{i_0 j'}$$

$$= \sum_{j} D'_{jj} \left(D'_{jj} - 2D_{i_0 i_0}\right) \mu_{i_0}^2 \theta_j^2 - \sum_{s=1}^{K} \theta_s \sum_{j=s} D'_{jj} \left(D'_{jj} - 2D_{i_0 i_0}\right) \mu_{i_0}^2$$

$$= \sum_{j} \mu_{i_0}^2 D'_{jj} \theta_j (\theta_j - 1) \left(D'_{jj} - 2D_{i_0 i_0}\right).$$

If (19) is satisfied, then $I \geq 0$ since $\theta_j < 1$. Additionally, it is straightforward to obtain $II = III = 0$. Thus, we can conclude that $\mathcal{E}_{D,D'}(\pi) - \sum_{s=1}^{K} \theta_s \mathcal{E}_{D,D'}(\hat{\pi}(s)) \geq 0$.

Condition (19) can be satisfied when $2\min_i D_{ii} \geq \max_i D_{ii}$. In fact, from (10), we have

$$D'_{jj} = D'_{jj}(t) = \frac{\sum_{i,i'} \pi(t-1)_{ij} D_{ii'} \pi(t-1)_{i'j}}{\sum_{i,i'} \pi(t-1)_{ij} \pi(t-1)_{i'j}} \leq \max(D),$$

in which $\max(D)$ must be achieved at the diagonal element for positive semidefinite matrix.

**Tensor formulation**  In numerical experiments, it is common to introduce a 4-way tensor:

$$\mathcal{L}(D, D')_{i,j,i',j'} \stackrel{\text{def.}}{=} \left(D_{ii'} - D'_{jj'}\right)^2,$$

then we have

$$\mathcal{L}(D, D') \otimes \pi \stackrel{\text{def.}}{=} \left(\sum_{i',j'} \mathcal{L}(D, D')_{i,j,i',j'} \pi_{i'j'}\right)_{i,j},$$

where $\otimes$ denotes the tensor-matrix multiplication. Subsequently, we can express the objective function in the the GW distance as follows,

$$\mathcal{E}_{D,D'}(\pi) = \langle \mathcal{L}(D, D') \otimes \pi, \pi \rangle,$$

where $\langle \cdot, \cdot \rangle$ denotes Frobenius product. And we can directly derive the derivative:

$$\frac{\nabla \mathcal{E}_{D,D'}(\pi)}{\nabla \pi} = 2\mathcal{L}(D, D') \otimes \pi.$$

**Numerical solver**  We will solve (12) by generating a sequence $\{\pi^{\tau,\varepsilon}\}$ that successively minimizing the approximate upper bound function of $f$:

$$
\begin{aligned}
\pi^{\tau+1,\varepsilon} &= \arg\min_{\pi \in \Pi(\frac{1_N}{N}, \cdot)} u(\pi, \pi^{\tau,\varepsilon}) \\
&= \arg\min_{\pi \in \Pi(\frac{1_N}{N}, \cdot)} f(\pi^{\tau,\varepsilon}) + \langle \nabla f(\pi^{\tau,\varepsilon}), \pi - \pi^{\tau,\varepsilon} \rangle + \varepsilon D_{\text{KL}}(\pi \| \pi^{\tau,\varepsilon}) \\
&= \arg\min_{\pi \in \Pi(\frac{1_N}{N}, \cdot)} \langle \nabla f(\pi^{\tau,\varepsilon}), \pi \rangle + \varepsilon D_{\text{KL}}(\pi \| \pi^{\tau,\varepsilon}) \\
&= \arg\min_{\pi \in \Pi(\frac{1_N}{N}, \cdot)} \langle \nabla f(\pi^{\tau,\varepsilon}), \pi \rangle + \varepsilon(\langle \pi, ln(\pi) \rangle - \langle \pi, ln(\pi^{\tau,\varepsilon}) \rangle) \\
&= \arg\min_{\pi \in \Pi(\frac{1_N}{N}, \cdot)} \langle \nabla f(\pi^{\tau,\varepsilon}) - \varepsilon ln(\pi^{\tau,\varepsilon}), \pi \rangle + \varepsilon \langle \pi, ln(\pi) \rangle,
\end{aligned}
$$

by defining $C' = \nabla f(\pi^{\tau,\varepsilon}) - \varepsilon ln(\pi^{\tau,\varepsilon})$, it is equivalent to solve

$$\pi^{\tau+1,\varepsilon} = \arg\min_{\pi \in \Pi(\frac{1_N}{N}, \cdot)} \langle C', \pi \rangle + \varepsilon \langle \pi, ln(\pi) \rangle.$$

This sub-problem can be reformulated as an Eulerian discrete Wasserstein Barycenter problem with an entropy regularizer, Introducing the dual variable $\psi \in \mathbb{R}^N$, the Lagarangian function is

$$\mathbf{L}(\pi, \psi) = \langle C', \pi \rangle + \varepsilon \langle \pi, ln(\pi) \rangle - \left\langle \psi, \pi \mathbf{1}_{k*1} - \frac{\mathbf{1}_{N*1}}{N} \right\rangle,$$

take first order gradient and we get

$$\frac{\partial \mathbf{L}(\pi, \psi)}{\partial \pi_{i,j}} = C'_{i,j} + \varepsilon \log(\pi_{i,j}) - \psi_i = 0,$$

$$\Rightarrow \pi_{i,j} = e^{\psi_i/\varepsilon} e^{-C'_{i,j}/\varepsilon} = u_i \mathbf{K}_{i,j}.$$

Besides, based on the constrain that:

$$\sum_{j=1}^k \pi_{ij}^{\tau+1,\varepsilon} = u_i \sum_{j=1}^k K_{ij} = \frac{1}{N},$$

we can get

$$u_i = \frac{1}{N \sum_{j=1}^k K_{ij}},$$

$$\pi_{i,j}^{\tau+1,\varepsilon} = \frac{K_{i,j}}{N \sum_{j=1}^k K_{ij}}.$$

It is evident that this is a linear programming problem, and as $\varepsilon$ approaches 0 the $\pi^{\tau+1,\varepsilon}$ solves an unconstrained strictly convex problem, which converges to $\pi^\star$ at a reasonable linear rateCominetti & Martín (1994), where $\pi^\star$ is the solution to the original Wasserstein barycenter problem $\min_{\pi \in \Pi(\frac{1_N}{N}, \cdot)} \langle C', \pi \rangle$ and is attained at an extremal point. In summary, the proposed learning algorithm is presented in Algorithm 1.

A.2   PROOFS OF THEOREMS

First we give two lemmas that will be used in the proofs.

**Lemma 1** *If $\lambda \geq N \max(D')^2 - \frac{\varepsilon N}{4}$, then we have*

$$u(\pi, \pi^{\tau,\varepsilon}) \geq f(\pi), \forall \pi, \pi^{\tau,\varepsilon} \in \Pi(\frac{\mathbf{1}_N}{N}, \cdot),$$

$$u(\pi^{\tau,\varepsilon}, \pi^{\tau,\varepsilon}) = f(\pi^{\tau,\varepsilon}), \forall \pi^{\tau,\varepsilon} \in \Pi(\frac{\mathbf{1}_N}{N}, \cdot).$$

This implies that the approximate function $u(\pi, \pi^{\tau,\varepsilon})$ is a tight upper bound of the original function $f(\pi)$.

**Proof** *Obviously $u(\pi, \pi^{\tau,\varepsilon})$ is continuously differentiable and $u(\pi^{\tau,\varepsilon}, \pi^{\tau,\varepsilon}) = f(\pi^{\tau,\varepsilon}), \forall \pi^{\tau,\varepsilon} \in \Pi(\frac{\mathbf{1}_N}{N}, \cdot)$, and we have*

$$\begin{aligned}
&u(\pi, \pi^{\tau,\varepsilon}) - f(\pi)\\
=&u(\pi, \pi^{\tau,\varepsilon}) - (\mathcal{E}_{D,D'}(\pi) - \lambda\langle\pi,\pi\rangle)\\
=&\mathcal{E}_{D,D'}(\pi^{\tau,\varepsilon}) - \lambda\langle\pi^{\tau,\varepsilon},\pi^{\tau,\varepsilon}\rangle + \langle\nabla\mathcal{E}_{D,D'}(\pi^{\tau,\varepsilon}) - 2\lambda\pi^{\tau,\varepsilon}, \pi - \pi^{\tau,\varepsilon}\rangle\\
&- \mathcal{E}_{D,D'}(\pi) + \lambda\langle\pi,\pi\rangle + \varepsilon D_{\mathrm{KL}}(\pi\|\pi^{\tau,\varepsilon})\\
=&\langle\mathcal{L}(D,D')\otimes\pi^{\tau,\varepsilon}, \pi^{\tau,\varepsilon}\rangle + \langle2\mathcal{L}(D,D')\otimes\pi^{\tau,\varepsilon}, \pi - \pi^{\tau,\varepsilon}\rangle - \langle\mathcal{L}(D,D')\otimes\pi,\pi\rangle\\
&- \lambda\langle\pi^{\tau,\varepsilon},\pi^{\tau,\varepsilon}\rangle - \langle2\lambda\pi^{\tau,\varepsilon}, \pi - \pi^{\tau,\varepsilon}\rangle + \lambda\langle\pi,\pi\rangle + \varepsilon D_{\mathrm{KL}}(\pi\|\pi^{\tau,\varepsilon})\\
=&\underbrace{-\langle\mathcal{L}(D,D')\otimes(\pi^{\tau,\varepsilon}-\pi), (\pi^{\tau,\varepsilon}-\pi)\rangle}_{I} + \underbrace{\lambda\langle\pi^{\tau,\varepsilon}-\pi, \pi^{\tau,\varepsilon}-\pi\rangle}_{II} + \underbrace{\varepsilon D_{\mathrm{KL}}(\pi\|\pi^{\tau,\varepsilon})}_{III}.
\end{aligned}$$

*We firstly introduce $T = \pi^{\tau,\varepsilon} - \pi$, and it satisfies $\sum_{j=1}^{K} T_{ij} = 0, for \forall i$. We can rewrite III as*

$$D_{\mathrm{KL}}(\pi\|\pi^{\tau,\varepsilon}) = \sum_{ij}\pi_{ij}ln(\frac{\pi_{ij}}{\pi_{ij}^{\tau,\varepsilon}}) = -2\sum_{ij}\pi_{ij}ln(\sqrt{\frac{\pi_{ij}^{\tau,\varepsilon}}{\pi_{ij}}}).$$

*Using $-ln(x) \leq 1 - x$ and $\sum_{ij}\pi_{ij} = \sum_{ij}\pi_{ij}^{\tau,\varepsilon} = 1$, we have*

$$\begin{aligned}
D_{\mathrm{KL}}(\pi\|\pi^{\tau,\varepsilon}) &\geq 2\sum_{ij}\pi_{ij}(1 - \sqrt{\frac{\pi_{ij}^{\tau,\varepsilon}}{\pi_{ij}}}) = \sum_{ij}(2\pi_{ij} - 2\sqrt{\pi_{ij}^{\tau,\varepsilon}\pi_{ij}})\\
&= \sum_{ij}(\pi_{ij} + \pi_{ij}^{\tau,\varepsilon} - 2\sqrt{\pi_{ij}^{\tau,\varepsilon}\pi_{ij}}) = \sum_{ij}(\sqrt{\pi_{ij}} - \sqrt{\pi_{ij}^{\tau,\varepsilon}})^2\\
&= \sum_{ij}(\frac{\pi_{ij} - \pi_{ij}^{\tau,\varepsilon}}{\sqrt{\pi_{ij}} + \sqrt{\pi_{ij}^{\tau,\varepsilon}}})^2 \geq \sum_{ij}(\frac{\pi_{ij} - \pi_{ij}^{\tau,\varepsilon}}{max(\sqrt{\pi_{ij}} + \sqrt{\pi_{ij}^{\tau,\varepsilon}})})^2.
\end{aligned}$$

*Since $\pi_{ij} \leq \frac{1}{N}$ for $\forall\pi \in \Pi(\frac{\mathbf{1}_N}{N}, \cdot)$, we have $(\sqrt{\pi_{ij}} + \sqrt{\pi_{ij}^{\tau,\varepsilon}}) \leq 2\sqrt{\frac{1}{N}}$, and*

$$III \geq \varepsilon\sum_{ij}(\frac{\pi_{ij} - \pi_{ij}^{\tau,\varepsilon}}{2\sqrt{\frac{1}{N}}})^2 = \frac{\varepsilon N}{4}\langle T, T\rangle. \tag{20}$$

*Therefore,*

$$u(\pi, \pi^{\tau,\varepsilon}) - f(\pi) \geq I + II + \frac{\varepsilon N}{4}\langle T, T\rangle. \tag{21}$$

*Notice that the negative I is the GW distance $\mathcal{E}_{D,D'}(T)$. Direct calculations lead to*

$$I = -\sum_{j}(D'_{jj})^2(\sum_{i}T_{ij})^2 + 2\|UTV^\top\|_F^2 \geq -\sum_{j}\hat{D}'(\sum_{i}T_{ij})^2, \tag{22}$$

*where we have introduced $\hat{D}' = \max(D'_{jj})^2$. Then by the Rearrangement Inequality*

$$A_n = \frac{(a_1 + a_2 + ... + a_n)}{n} \leq \sqrt{\frac{a_1^2 + a_2^2 + ... + a_n^2}{n}} = Q_n,$$
$$-(a_1 + a_2 + ... + a_n)^2 \geq -n * (a_1^2 + a_2^2 + ... + a_n^2),$$

*we can get*

$$-\sum_j \hat{D}' (\sum_i T_{ij})^2 \geq -\sum_j \hat{D}' N \sum_i T_{ij}^2 = -N\hat{D}' \langle T, T \rangle. \tag{23}$$

*Combining Equations(21) , (22), and (23), we can obtain:*

$$u(\pi, \pi^{\tau,\varepsilon}) - f(\pi) \geq (\lambda - N D' + \frac{\overset{\wedge}{\varepsilon N}}{4}) \langle T, T \rangle, \tag{24}$$

*which leads to the conclusion that $u(\pi, \pi^{\tau,\varepsilon}) - f(\pi) \geq 0$ if $\lambda \geq N\hat{D}' - \frac{\varepsilon N}{4}$.*

**Lemma 2** *If $\lambda \geq N \times max(D')^2 - \frac{\varepsilon N}{4}$, then we have*

$$u'(\pi, \pi^{\tau,\varepsilon}; d)|_{\pi=\pi^{\tau,\varepsilon}} = f'(\pi; d), \quad \forall d \text{ with } \pi + d \in \Pi(\frac{\mathbf{1}_N}{N}, \cdot).$$

This guarantees that the first order behavior of $u(\pi, \pi^{\tau,\varepsilon})$ is the same as $f(\pi)$ locally (note that the directional derivative $u'(\pi, \pi^{\tau,\varepsilon}; d)$ is only with respect to the variable $\pi$).

**Proof** *Assume to the contrary that there exist a transport plan $\pi^{\tau,\varepsilon} \in \Pi(\frac{\mathbf{1}_N}{N}, \cdot)$ and a direction $d$ such that*

$$u'(\pi, \pi^{\tau,\varepsilon}; d)|_{\pi=\pi^{\tau,\varepsilon}} \neq f'(\pi^{\tau,\varepsilon}; d).$$

*Noticing that $\pi^{\tau,\varepsilon}$ may be at the boundary of $\Pi(\frac{\mathbf{1}_N}{N}, \cdot)$ such that $\pi^{\tau,\varepsilon} - \alpha d$ does not belong to $\Pi(\frac{\mathbf{1}_N}{N}, \cdot)$ for $\forall \alpha > 0$, we can introduce $\bar{\pi} = \pi^{\tau,\varepsilon} + \alpha d$ for some $\alpha > 0$. Due to the fact that $f(\pi)$ and $u(\pi, \pi^{\tau,\varepsilon})$ are continuously differentiable, $\bar{\pi}$ must belong to the interior of $\Pi(\frac{\mathbf{1}_N}{N}, \cdot)$ and satisfy*

$$u'(\pi, \bar{\pi}; d)|_{\pi=\bar{\pi}} \neq f'(\bar{\pi}; d).$$

*However, Lemma 1 implies that*

$$u'(\pi, \bar{\pi}; d)|_{\pi=\bar{\pi}} = \lim_{\lambda \downarrow 0} \frac{u(\bar{\pi} + \lambda d, \bar{\pi}) - u(\bar{\pi}, \bar{\pi})}{\lambda}$$
$$\geq \lim_{\lambda \downarrow 0} \frac{f(\bar{\pi} + \lambda d) - f(\bar{\pi})}{\lambda} = f'(\bar{\pi}; d),$$
$$u'(\pi, \bar{\pi}; d)|_{\pi=\bar{\pi}} = \lim_{\lambda \downarrow 0} \frac{u(\bar{\pi}, \bar{\pi}) - u(\bar{\pi} - \lambda d, \bar{\pi})}{\lambda}$$
$$\leq \lim_{\lambda \downarrow 0} \frac{f(\bar{\pi}) - f(\bar{\pi} - \lambda d)}{\lambda} = f'(\bar{\pi}; d),$$

*which implies that $u'(\pi, \bar{\pi}; d)|_{\pi=\bar{\pi}} = f'(\bar{\pi}; d)$, leading to the contradiction.*

**Proof of Theorem 1** Firstly, by Lemma 1 and $\pi^{\tau+1,\varepsilon} = \arg \min_{\pi \in \Pi(\frac{\mathbf{1}_N}{N}, \cdot)} u(\pi, \pi^{\tau,\varepsilon})$, we have

$$f\left(\pi^{\tau+1,\varepsilon}\right) \leq u\left(\pi^{\tau+1,\varepsilon}, \pi^{\tau,\varepsilon}\right) \leq u\left(\pi^{\tau,\varepsilon}, \pi^{\tau,\varepsilon}\right) = f\left(\pi^{\tau,\varepsilon}\right), \quad \forall \tau = 0, 1, 2, \ldots \tag{25}$$

So the sequence of the objective function values are non-increasing, i.e.

$$f\left(\pi^{0,\varepsilon}\right) \geq f\left(\pi^{1,\varepsilon}\right) \geq f\left(\pi^{2,\varepsilon}\right) \geq \ldots$$

Assuming that there exists a subsequence $\{\pi^{r_j,\varepsilon}\}$ converging to a limit point $\pi(t+1)$, then for $\forall \pi \in \Pi(\frac{\mathbf{1}_N}{N}, \cdot)$, we have

$$u\left(\pi^{r_{j+1},\varepsilon}, \pi^{r_{j+1},\varepsilon}\right) = f\left(\pi^{r_{j+1},\varepsilon}\right) \leq f\left(\pi^{r_j+1,\varepsilon}\right) \leq u\left(\pi^{r_j+1,\varepsilon}, \pi^{r_j,\varepsilon}\right) \leq u\left(\pi, \pi^{r_j,\varepsilon}\right).$$

Letting $j \to \infty$, we obtain

$$u(\pi(t+1), \pi(t+1)) \leq u(\pi, \pi(t+1)), \quad \forall \pi \in \Pi(\frac{\mathbf{1}_N}{N}, \cdot),$$

which implies

$$u'(\pi, \pi(t+1); d)|_{\pi=\pi(t+1)} \geq 0, \quad \forall d \quad \text{s.t. } \pi(t+1) + d \in \Pi(\frac{\mathbf{1}_N}{N}, \cdot).$$

This together with Lemma 2 implies that $\pi(t+1)$ is a stationary point of $f(\pi)$.

**Proof of Theorem 3** According to the Theorem 1, at the $t$-th outer loop:
$$\pi^{0,\varepsilon} = \pi(t),$$
$$\pi^{\tau+1,\varepsilon} = \arg \min_{\pi \in \Pi(\frac{\mathbf{1}_N}{N}, \cdot)} u(\pi, \pi^{\tau,\varepsilon}), \quad \tau = 0, 1, 2, \dots$$
$$\lim_{\tau \to \infty} \pi^{\tau,\varepsilon} = \pi(t+1).$$

Based on the monotonicity of the objective function (25) and Lemma 1, we can conclude that

$$f(\pi(t)) - f(\pi(t+1))$$
$$= \sum_{\tau=0}^{\infty} f(\pi^{\tau,\varepsilon}) - f(\pi^{\tau+1,\varepsilon})$$
$$\geq \sum_{\tau=0}^{\infty} u(\pi^{\tau+1,\varepsilon}, \pi^{\tau,\varepsilon}) - f(\pi^{\tau+1,\varepsilon})$$
$$\geq \sum_{\tau=0}^{\infty} (\lambda - N\hat{D}' + \frac{\varepsilon N}{4}) \|\pi^{\tau,\varepsilon} - \pi^{\tau+1,\varepsilon}\|_F^2,$$

where the last inequality is a consequence of Equation (24). Since $f(\cdot)$ attains minimum at the extremal point, we have that $\langle \pi(t), \pi(t) \rangle = \langle \pi(t+1), \pi(t+1) \rangle = \frac{1}{N}$ is a constant, which implies $f(\pi(t)) = \mathcal{E}_{D,D'(t)}(\pi(t)) - \frac{\lambda}{N}$, $f(\pi(t+1)) = \mathcal{E}_{D,D'(t)}(\pi(t+1)) - \frac{\lambda}{N}$, and $\sum_\tau \|\pi^{\tau,\varepsilon} - \pi^{\tau+1,\varepsilon}\|_F^2$ is bounded by a positive multiple of $\|\mathcal{E}_{D,D'(t)}(\pi(t+1)) - \mathcal{E}_{D,D'(t)}(\pi(t))\|_F$.

Moreover, by the monotonicity and lower boundedness of the objective function $f$, $\lim_{t \to \infty} f(\pi(t)) - f(\pi(t+1)) = 0$, then for sufficiently large $t$, we can infer that

$$\sum_{\tau=0}^{\infty} \|\pi^{\tau,\varepsilon} - \pi^{\tau+1,\varepsilon}\|_F^2 < \frac{1}{N^2}.$$

When $\varepsilon = 0$, $\pi^{\tau,0}$ belongs to $\frac{1}{N}\text{Ind}^{N \times K}$ for $\forall \tau$. This implies $\{\pi^{\tau,0}\}$ must be a constant sequence and $\pi(t) = \pi(t+1)$. When $\varepsilon > 0$, for sufficiently large $t$ and the linear convergence of sinkhorn algorithm, we have

$$\|\pi(t) - \pi(t+1)\|_F^2 \leq 2\{\|\pi(t) - \pi^{T,\varepsilon}\|_F^2 + \|\pi^{T,\varepsilon} - \pi(t+1)\|_F^2\}$$
$$\leq 2\{T \sum_{\tau=0}^{T-1} \|\pi^{\tau,\varepsilon} - \pi^{\tau+1,\varepsilon}\|_F^2 + \|\pi^{T,\varepsilon} - \pi(t+1)\|_F^2\}$$
$$< \frac{1}{N^2}.$$

This also implies $\pi(t) = \pi(t+1)$.

# B MORE EXPERIMENTAL RESULTS

## B.1 DEGENERATED MODEL

When $\mu$ is a uniform distribution, optimizing the mapping $\Gamma$ is equivalent to finding the optimal $T = N\Gamma \in Ind^{N*K}$, where $Ind^{N*K}$ is the set of indicator matrix (rectangular permuation matrix),

so $T_{ij}T_{i'j} = 1 \iff x_i, x_{i'} \in V_j$ and $\sum_{i,i'} T_{ij}T_{i'j} = |V_j|^2$. By removing the constant term, the objective function in GW discrepancy can be reformulated as:

$$
\begin{aligned}
\mathcal{E}_{D,D'}(T) &= \sum_{i,i'} \sum_{j,j'} \left( D_{i,i'} - D'_{j,j'} \right)^2 T_{ij}T_{i'j'} \\
&= \sum_{i,i'} \sum_{j,j'} {D'_{j,j'}}^2 T_{ij}T_{i'j'} - 2 \sum_{i,i'} \sum_{j,j'} T_{ij} D_{i,i'} T_{i'j'} D'_{j,j'} \\
&= \sum_j {D'_{j,j}}^2 \sum_{i,i'} T_{ij}T_{i'j} - 2 \sum_{i,i'} \sum_{j,j'} T_{ij} D_{i,i'} T_{i'j'} D'_{j,j'} \\
&= -2Tr(T^\top DTD') + Tr(D'^2 T^\top \mathbf{1}\mathbf{1}^\top T).
\end{aligned}
$$

So the barycenter problem (6) can be rewritten as

$$
\min_{D' \in Diag_+} \min_{T \in Ind^{N*K}} -2Tr(T^\top DTD') + Tr(D'^2 T^\top \mathbf{1}\mathbf{1}^\top T). \tag{26}
$$

We will show the connections between EGWB and other methods in following sections.

**GWL and SpecGWL**   Clearly, if we fix $\nu$ and set $D' = \text{diag}(\nu)$, while also setting $\lambda = 0$, our model degenerates into GWL or SpecGWL (depending on whether we choose the adjacency matrix or the heat kernel matrix for $D$).

**srGW**   If we fixed $D' = \mathbb{I}_K$, problem (8) then reduces to the scenario of Semi-relaxed Gromov-Wasserstein graph partitioning (Vincent-Cuaz et al., 2022). Indeed, the objective function (26) becomes

$$
\min_{T \in Ind^{N*K}} -2Tr(T^\top DT) + Tr(T^\top \mathbf{1}\mathbf{1}^\top T), \tag{27}
$$

if $D$ is the adjacency matrix, the first term in Equation (30) is exactly the original Min-Cut problem. The second term is exactly the exclusive lasso Zhou et al. (2010) defined as follow,

$$
\|T\|_e = \sum_{j=1}^K \left( \sum_{i=1}^N \|T_{ij}\| \right)^2 = Tr(T^\top \mathbf{1}\mathbf{1}^\top T),
$$

it consists of $l_1$-norm for within-group sparsity and $l_2$-norm for between-group non-sparseness. Intuitively speaking, exclusive lasso makes variables in the same group compete with each other in each group. According to the Cauchy inequality, we have

$$
\sum_{j=1}^K \left( \sum_{i=1}^N \|T_{ij}\| \right)^2 \geq \frac{\left( \sum_{j=1}^K \sum_{i=1}^N \|T_{ij}\| \right)^2}{K} = \frac{N^2}{K},
$$

which arrives at its minimum if and only if $|V_j| = \sum_{i=1}^N T_{ij} = \frac{N}{K}, \forall j = 1, \cdots, K$, which means each cluster has the same number of data points. Adding this regularization can also avoid the situation that a certain class contains only a few points to a certain extent. It is noted that srGW can be viewed as a continuous optimization problem for this via the Kantorovich relaxation of (27).

**Balanced Min-Cut**   Taking a step further, if we posit the form of $D' = s\mathbb{I}_K$, where $s$ is a variable, it becomes balanced Min-Cut problem(Chen et al., 2017):

$$
\min_s \min_{T \in Ind^{N*K}} -2sTr(T^\top DT) + s^2 Tr(T^\top \mathbf{1}\mathbf{1}^\top T). \tag{28}
$$

We can find the optimal value by alternate optimization.

(1) Obviously, when $T$ is fixed, the objective function in Equation (28) is a convex quadratic function about s, so we have

$$
s^* = \frac{Tr(T^\top DT)}{Tr(T^\top \mathbf{1}\mathbf{1}^\top T)}, \tag{29}
$$

the scalar $s$ is learned to balance the partition across all clusters.

(2) When $s$ is fixed, it is equivalent to solve

$$\min_{T \in Ind^{N*K}} -2Tr(T^\top DT) + sTr(T^\top \mathbf{1}\mathbf{1}^\top T), \tag{30}$$

Upon obtaining the optimal solution, substituting $s^*$ from Equation (29) into Equation (28) yields:

$$\min_s \min_{T \in Ind^{N*K}} s^* * (-Tr(T^\top DT)),$$

where $s$ exhibits inverse proportionality with $\|T\|e$. The second term corresponds to the objective function of classical Min-Cut and is less than zero. Thus, we aim to find the maximum possible value of $s$, which corresponds to minimizing the magnitude of $\|T\|_e$. Therefore, the entire model can perform dual tasks of minimizing the graph cut and promoting balance across all clusters.

**NMF form of Balanced Min-Cut**   Clearly for any indicator matrix $T$ we have $Tr\left(TT^\top TT^\top\right) = \sum_{j=1}^{K}\left(\sum_{i=1}^{N} T_{ij}\right)^2$, and we know $Tr\left(TT^\top TT^\top\right) = Tr\left(T^\top \mathbf{1}\mathbf{1}^\top T\right) = \|T\|_e$, so the objective function in Equation (28) can be reformulated as a NMF problem:

$$\min_s \min_{T \in Ind^{N*K}} -2sTr(T^\top DT) + s^2 Tr(T^\top \mathbf{1}\mathbf{1}^\top T)$$

$$= \min_s \min_{T \in Ind^{N*K}} -2sTr(T^\top DT) + s^2 Tr\left(TT^\top TT^\top\right)$$

$$\iff \min_s \min_{T \in Ind^{N*K}} Tr(DD^\top) - 2sTr(T^\top DT) + s^2 Tr\left(TT^\top TT^\top\right)$$

$$= \min_s \min_{T \in Ind^{N*K}} \|D - sTT^\top\|_F^2.$$

**Enhanced balanced Min-Cut (EBMC)**   It's worth noting that data with a perfect cluster structure (where each class has an equal number of data points) is uncommon in real-life scenarios. The assumption in srGW and balanced Min-Cut that the diagonal elements of $D'$ are equal is insufficient for distinguishing dissimilarities among different clusters. Therefore, our algorithm treats $D'$ as variables for optimization, and the objective function (26) of EGWB is equivalent to enhanced balanced Min-Cut (Chen et al., 2020):

$$\max_{D' \in Diag_+, T \in Ind^{N*K}} \sum_{l=1}^{K} T_{:,l}^\top (2D_{ll}' D - (D_{ll}')^2 \mathbf{1}\mathbf{1}^\top) T_{:,l}. \tag{31}$$

In particular, when we substitute the optimal $D'$ obtained in the previous step (10) into the subproblem for $\pi$, we can obtain:

$$\pi^* = \arg\min_\pi \sum_{l=1}^{K} -\frac{(\pi_{:,l}^\top D \pi_{:,l})^2}{|V_l|^2}. \tag{32}$$

Recall that the classical normalized cut problem can be formulated as:

$$\min_T \sum_{l=1}^{K} \frac{T_{:,l}^\top L_W T_{:,l}}{T_{:,l}^\top D_W T_{:,l}} \iff \min_T \sum_{l=1}^{K} -\frac{T_{:,l}^\top W T_{:,l}}{T_{:,l}^\top D_W T_{:,l}},$$

where $W$ is the adjacency matrix, and $L_W$ and $D_W$ are the corresponding graph laplacian and degree matrices. Here, the term $T_{:,l}^\top D_W T_{:,l}$ is included to improve the robustness of the model when dealing with isolated nodes (Shi & Malik, 2000). It can be observed that (32) has a similar objective of maximizing the within-cluster similarities, but utilize different normalization terms.

**Weighted symmetric NMF form of EBMC**   Clearly for any indicator matrix $T$ we have $T_{ij}T_{ij'} = \delta_{jj'}$, and

$$Tr(TD'T^\top TD'T^\top) = \sum_{i,t,k,l,m,n} T_{ik}D_{kl}'T_{tl}T_{tm}D_{mn}'T_{in}$$

$$= \sum_{i,t,k=n,l=m} T_{ik}T_{tl}D_{kl}'D_{kl}' = \sum_{j,j'}\sum_{i,i'} D_{j,j'}'^2 T_{ij}T_{i'j'},$$

so Equation (26) can be reformulated as

$$\min_{D'} \min_{T \in Ind^{N*K}} \sum_{i,i'} \sum_{j,j'} D'^2_{j,j'} T_{ij} T_{i'j'} - 2 \sum_{i,i'} \sum_{j,j'} T_{ij} D_{i,i'} T_{i'j'} D'_{j,j'}$$

$$\iff \min_{D'} \min_{T \in Ind^{N*K}} Tr(TD'T^\top TD'T^\top) - 2Tr(T^\top D T D')$$

$$\iff \min_{D'} \min_{T \in Ind^{N*K}} \|D - TD'T^\top\|_F^2,$$

which is kind of like weighted symmetric NMF $D \approx TD'T^\top$. We can employ the Lagrangian function to obtain the stationary equation and the KKT complementarity slackness condition at the stationary point,

$$L(T, D', \phi, \psi) = \|D - TD'T^\top\|_F^2 - \phi T - \psi D',$$

$$\frac{\partial L}{\partial T} = -4DTD' + 4TD'T^\top TD' - \phi,$$

$$\frac{\partial L}{\partial D'} = 2T^\top DT + 2T^\top TD'T^\top T - \psi,$$

$$\frac{\partial L}{\partial T^*} = \frac{\partial L}{\partial T^*} * T^* = 0 = \frac{\partial L}{\partial (D')^*} * (D')^* = \frac{\partial L}{\partial (D')^*},$$

$$\phi^* * T^* = 0 = \psi^* * (D')^*,$$

subsequently, we can employ methods such as gradient descent to find the fixed point $(D')^*$ and $T^*$.

**Max-Dicut** When considering the partition of a directed graph, such as the oil trade between countries, we are more concerned with the relationships between different clusters. In this case, the Max-Dicut algorithm is commonly used (Goemans & Williamson, 1995). Correspondingly, we only need to utilize the upper triangular matrix $D'$ with a main diagonal of zeros to represent the graph $G'$, rather than a diagonal matrix for disconnected graph. It should be noted that the cut between two clusters can still be defined as

$$\text{cut}(V_i, V_j) = \pi_{:,i}^\top D \pi_{:,j}.$$

Similarly, substituting the optimal $D'$ obtained in the previous step (10) into the subproblem for $\pi$ yields

$$\pi^* = \arg\max_\pi \sum_{i=1}^K \sum_{j=i+1}^K \frac{\text{cut}(V_i, V_j)^2}{|V_i||V_j|} \tag{33}$$

$$= \arg\max_\pi \sum_{i=1}^K \sum_{j=i+1}^K \underbrace{\frac{\text{cut}(V_i, V_j)}{|V_i| + |V_j|}}_{I} \times \underbrace{\left( \frac{\text{cut}(V_i, V_j)}{|V_i|} + \frac{\text{cut}(V_i, V_j)}{|V_j|} \right)}_{II}. \tag{34}$$

where $I$ and $II$ can be regarded as a variant of the average cut. Furthermore, if the data exhibits an obvious flow structure, inspired by the work in (Laenen & Sun, 2020), we can define the flow ratio by setting $D'$ as an upper triangular matrix with only the sub-diagonal elements being non-zero. This is because even though most of the other terms are small, their sum can still dominate the entire objective function. Therefore, we can consider the following max flow ratio problem:

$$\pi^* = \arg\max_\pi \sum_{i=1}^K \frac{\text{cut}(V_i, V_{i+1})^2}{|V_i||V_{i+1}|}. \tag{35}$$

**Relaxation of these models** For such a discrete optimization problem (26) in the indicator matrix space, we can solve it using a new relaxation techniques. For example, in the context of NMF theory, we can retain nonnegativity and relax orthogonality to obtain $\pi = \arg\min_{\pi_{ij} \geq 0} \|D - \pi D' \pi^\top\|_F^2$ in the form of weighted symmetric NMF. Alternatively, in the framework of spectral clustering theory, we can relax the nonnegativity and retain the orthogonality, resulting in a generalized eigenvector problem. Hereby, from the perspective of OT and using the connections above, we can use Kantorovich's relaxation to solve these problems in the probability space for $\pi$.

### B.2 MORE EXPERIMENTAL RESULTS

#### B.2.1 SYNTHETIC DATA

We first demonstrate the advantages of our algorithm over GWL and SpecGWL using simple synthetic datasets.

**Datasets and methods** We first consider the implementation of the $datasets.make\_blobs$ function in the $Scikit - learn$ library to construct four blobs, each containing 100, 200, 300, and 400 points respectively. The standard deviation of the clusters is set to 0.55, making it a simple toy dataset that can be perfectly separated using K-means or spectral clustering, see Figure 4a. Then we change standard deviation of the clusters to 0.55, 1.0, 1.5, and 2.0, respectively, as shown in Figure 2a.

**Experimental settings** We use $k = 200$ to construct the Gaussian kernel adjacency matrix and its corresponding graph Laplacian matrix and use $t = 100$ to construct the heat kernel matrix in SpecGWL. Recall a pre-estimated cluster size $\hat{\nu}$ is needed in the framework of GWL and SpecGWL. Here we employ two different estimates: $\hat{\nu} \approx (0.25, 0.25, 0.25, 0.25)$ given in (Xu et al., 2019a) and the truth $\nu_{true} = (0.1, 0.2, 0.3, 0.4)$. The results are listed in Figure 4 and Figure 2.

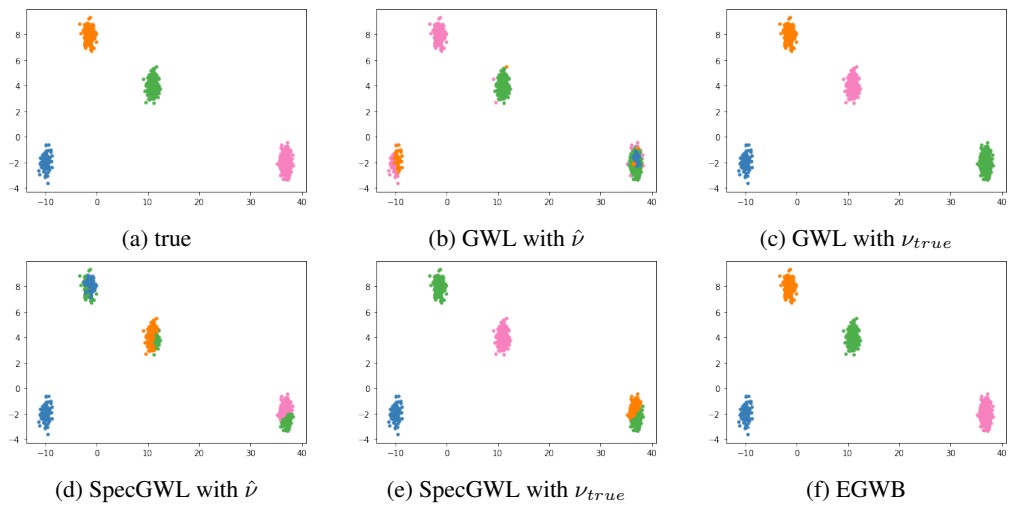

(a) true          (b) GWL with $\hat{\nu}$          (c) GWL with $\nu_{true}$

(d) SpecGWL with $\hat{\nu}$          (e) SpecGWL with $\nu_{true}$          (f) EGWB

Figure 4: Clustering results. From left to right, 4 blobs with same standard deviation 0.55, there are 100, 200, 300, and 400 points respectively. Different colors represent different clusters.

**Results and discussion** Note that by using heat kernel instead of adjacency matrix, SpecGWL performs better than GWL. The quality of the estimated hyperparameter $\hat{\nu}$ directly affects the clustering results of GWL and SpecGWL algorithms. The more accurate the estimation of $\hat{\nu}$, the better the clustering outcomes. Even when using the truth $\nu_{true}$, neither GWL nor SpecGWL can obtain correct clustering results (see Figure 2c and Figure 2e), while EGWB can lead to correct results without any prior information (see Figure 4f and Figure 2f).

#### B.2.2 REAL DATA

When dealing with real-life data, we employed the algorithms $greedy\_modularity\_communities$ and $asyn\_fluidc$ from the $networkx.algorithms.community$ library, as well as the $Infomap$ algorithm from the $infomap$ library, for graph partitioning methods Newman Fast Algorithm, FluidC, and Infomap, respectively. We utilized the default parameters for these algorithms. When applying GWL and SpecGWL, we used standard projected gradient descent and hand-tuned the required hyperparameters $t_{opt}$ and $\varepsilon$. We employed four common clustering evaluation metrics to assess the effectiveness of our algorithm: Adjusted Rand Index (ARI) (Yeung & Ruzzo, 2001), V-measure (Rosenberg & Hirschberg, 2007), Fowlkes-Mallows Index (FMI) (Fowlkes & Mallows, 1983), and Adjusted Mutual Information (AMI) (Vinh et al., 2009). In order to maintain consistency with other

GW-based methods, for AMI, we utilized both 'arithmetic' and 'max' averaging methods, denoted as AMI and $AMI_{max}$ respectively. The results are presented in Tables 3, 4, 5, 1 and 6

Table 3: Comparison of $ARI$ across a variety of datasets

| | Wikipedia | | EU-email | | Amazon | | Village | |
|---|---|---|---|---|---|---|---|---|
| | noisy | raw | noisy | raw | noisy | raw | noisy | raw |
| EGWB (ours) | 0.397 | 0.436 | 0.502 | 0.522 | 0.317 | 0.663 | 0.886 | 0.904 |
| GWL | 0.160 | 0.217 | 0.229 | 0.194 | 0.154 | 0.186 | 0.374 | 0.530 |
| SpecGWL | 0.267 | 0.298 | 0.262 | 0.302 | 0.172 | 0.525 | 0.660 | 0.744 |
| srGW | 0.401 | 0.441 | 0.433 | 0.501 | 0.286 | 0.548 | 0.691 | 0.782 |
| Fluid | 0.243 | NA | 0.198 | NA | 0.355 | NA | 0.372 | NA |
| Newman | 0.487 | 0.556 | 0.154 | 0.172 | 0.371 | 0.458 | 0.701 | 0.829 |
| InfoMap | 0.448 | 0.445 | 0.163 | 0.315 | 0.241 | 0.897 | 0.131 | 0.829 |

Table 4: Comparison of $V - measure$ across a variety of datasets

| | Wikipedia | | EU-email | | Amazon | | Village | |
|---|---|---|---|---|---|---|---|---|
| | noisy | raw | noisy | raw | noisy | raw | noisy | raw |
| EGWB (ours) | 0.553 | 0.627 | 0.642 | 0.687 | 0.735 | 0.824 | 0.883 | 0.921 |
| GWL | 0.361 | 0.489 | 0.507 | 0.532 | 0.340 | 0.381 | 0.522 | 0.698 |
| SpecGWL | 0.488 | 0.571 | 0.502 | 0.625 | 0.532 | 0.733 | 0.745 | 0.832 |
| srGW | 0.597 | 0.643 | 0.585 | 0.643 | 0.610 | 0.755 | 0.823 | 0.872 |
| Fluid | 0.322 | NA | 0.490 | NA | 0.203 | NA | 0.510 | NA |
| Newman | 0.610 | 0.669 | 0.353 | 0.478 | 0.680 | 0.780 | 0.727 | 0.840 |
| InfoMap | 0.580 | 0.662 | 0.511 | 0.650 | 0.527 | 0.944 | 0.215 | 0.840 |

Table 5: Comparison of $FMI$ across a variety of datasets

| | Wikipedia | | EU-email | | Amazon | | Village | |
|---|---|---|---|---|---|---|---|---|
| | noisy | raw | noisy | raw | noisy | raw | noisy | raw |
| EGWB (ours) | 0.433 | 0.475 | 0.503 | 0.548 | 0 .635 | 0.713 | 0.830 | 0.913 |
| GWL | 0.262 | 0.320 | 0.267 | 0.231 | 0.247 | 0.277 | 0.430 | 0.573 |
| SpecGWL | 0.403 | 0.399 | 0.324 | 0.344 | 0.403 | 0.589 | 0.735 | 0.768 |
| srGW | 0.453 | 0.456 | 0.501 | 0.528 | 0.563 | 0.610 | 0.802 | 0.822 |
| Fluid | 0.443 | NA | 0.198 | NA | 0.435 | NA | 0.677 | NA |
| Newman | 0.528 | 0.633 | 0.221 | 0.289 | 0.452 | 0.569 | 0.715 | 0.846 |
| InfoMap | 0.487 | 0.481 | 0.290 | 0.377 | 0.383 | 0.911 | 0.215 | 0.846 |

When employing our EGWB algorithm, we use a linear combination of the results from GWL, SpecGWL, and the joint distribution as initial values. A detailed analysis is provided in the next section. Both of them utilized the heat kernel generated in SpecGWL as the distance matrix $D$. For the hyperparameter $\lambda$, we apply a continuation scheme $\lambda = \max(\left(\frac{t}{\lambda_a}\right)^{\lambda_b}, 1) \times \left(N \times \max(D')^2 - \frac{\varepsilon N}{4}\right)$, where $t$ is the number of iteration, the hyperparameters $\lambda_a = 20$ and $\lambda_b = 5$ are set based on large experiments. The remaining performance metrics are displayed in Tables 3, 4, 5 and 6.

## B.3 SENSITIVITY ANALYSIS OF ALGORITHM PARAMETERS

In this section, we conduct a sensitivity analysis of our algorithm by varying its parameters and observing the impact on performance. We utilize the Indian village dataset as a representative example, and similar results are observed across all datasets. The key algorithmic parameters investigated are as follows:

### B.3.1 INITIAL TRANSPORT PLAN $\pi_0$

**Initialization method** Typically, clustering algorithms based on Gromov-Wasserstein distance initiate with the joint distribution of source distribution $p_s$ and target distribution $p_t$ as the algorithm's

Table 6: Comparison of $AMI_{max}$ across a variety of datasets

|  | Wikipedia | | EU-email | | Amazon | | Village | |
| --- | --- | --- | --- | --- | --- | --- | --- | --- |
|  | noisy | raw | noisy | raw | noisy | raw | noisy | raw |
| EGWB (ours) | 0.534 | 0.564 | 0.595 | 0.601 | 0.744 | 0.767 | 0.886 | 0.922 |
| GWL | 0.315 | 0.436 | 0.358 | 0.412 | 0.306 | 0.343 | 0.510 | 0.516 |
| SpecGWL | 0.388 | 0.507 | 0.412 | 0.493 | 0.501 | 0.675 | 0.733 | 0.821 |
| srGW | 0.556 | 0.576 | 0.560 | 0.557 | 0.533 | 0.696 | 0.802 | 0.850 |
| Fluid | 0.289 | NA | 0.395 | NA | 0.205 | NA | 0.435 | NA |
| Newman | 0.341 | 0.382 | 0.231 | 0.312 | 0.668 | 0.772 | 0.721 | 0.880 |
| InfoMap | 0.329 | 0.377 | 0.350 | 0.374 | 0.518 | 0.942 | 0.162 | 0.880 |

initial value, denoted as $\pi_0 = p_s \times p_t^T$. However, in our algorithm, we will calculate $p_t$ using $\pi$, rendering $p_t$ an unknown variable. We may assume $p_t$ to be a uniform distribution. Alternatively, we can employ the outcomes of GWL and SpecGWL, denoted as $\pi_{GWL}$ and $\pi_{SpecGWL}$ respectively, as prior expert knowledge. We construct

$$\hat{\pi}_0 = w * \pi_{GWL} + (1 - w) * \pi_{SpecGWL},$$

from which we compute $\hat{p}_t$. Finally, we set

$$\pi_0 = \rho * \hat{\pi}_0 + (1 - \rho) * (p_s \times \hat{p}_t^T)$$

as the initial value for the process (9).

Table 7: Different setup for initial $\pi_0$

|  | $w$ | $\rho$ |
| --- | --- | --- |
| $\pi_{01}$ | 1 | 0 |
| $\pi_{02}$ | 1 | 0.5 |
| $\pi_{03}$ | 0 | 0 |
| $\pi_{04}$ | 0 | 0.5 |
| $\pi_{05}$ | 0.5 | 0 |
| $\pi_{06}$ | 0.5 | 0.5 |

**Construction** Subsequently, we have devised six distinct initial guesses for the transport plan $\pi_0$, and their comprehensive configurations are outlined in Table 7. These configurations correspond to various combinations of GWL and SpecGWL, along with the inclusion of the joint distribution.

Following this, we conducted experiments under the scenario where $D = Heat[t]$ and $\mu$ is uniformly distributed. Figure 5 displays the variations in five different metrics for these six different initial values. We set $t$ to have 100 evenly spaced points in the interval $[1, 60]$. This analysis reveals a consistent trend across all evaluation metrics.

Henceforth, our focus will be directed towards the individual analysis of a single specific evaluation metric.

**Ablation study of** $w$ By computing the ARI of EGWB under the conditions of $\rho = 0.5$ and $\rho = 0$, the results depicted in Figure 6 reveal that as long as the initial distribution incorporates the results from SpecGWL, the final convergence outcomes tend to be better. Therefore, in the subsequent experiments, we adopt $w = 0.5$, which encompasses information from both prior expert insights, rendering the algorithm more robust.

**Ablation study of** $\rho$ By computing the ARI of EGWB under the conditions of $w = 0, 0.5$ or $1$, the results depicted in Figure 7 , which highlights that incorporating the joint distribution into the initial guess (i.e., $\rho = 0.5$) yields superior results. Additionally, Figure 7a demonstrates the robustness of our algorithm to variations in the heat kernel parameter $t$ when the initial values are fixed. Figure 7b suggests that using SpecGWL as an initial guess offers little improvement in AMI, possibly due to prematurely entering a stable state. Lastly, Figure 7c indicates that employing $\pi_{06}$ as an initial guess results in the most favorable outcomes.

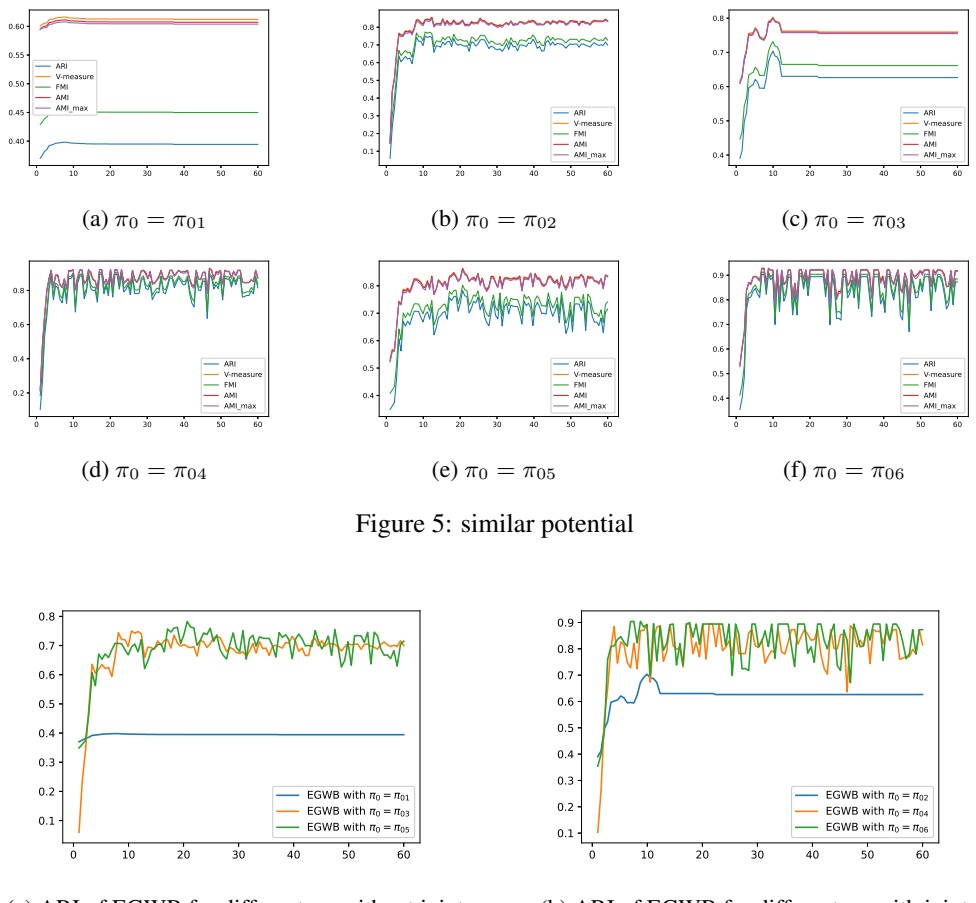

Figure 5: similar potential

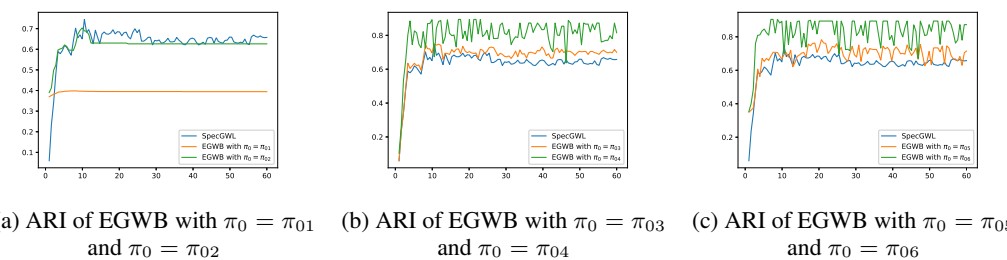

(a) ARI of EGWB for different $\pi_0$ without joint distribution

(b) ARI of EGWB for different $\pi_0$ with joint distribution

Figure 6: In typical scenarios, having a value of $w > 0$ serves as a preferable initial guess. This is because it incorporates information from SpecGWL, which tends to lead to a better local minimum compared to GWL alone.

We commence our experimentation by varying $\rho$ and observing the outcomes for a fixed t (thus fixed $D = Heat[t]$), as depicted in Figure 8. It becomes evident that our algorithm exhibits robustness concerning different values of $\rho$. Notably, when $\rho \in (0, 1)$, the results remain relatively consistent.

(a) ARI of EGWB with $\pi_0 = \pi_{01}$ and $\pi_0 = \pi_{02}$

(b) ARI of EGWB with $\pi_0 = \pi_{03}$ and $\pi_0 = \pi_{04}$

(c) ARI of EGWB with $\pi_0 = \pi_{05}$ and $\pi_0 = \pi_{06}$

Figure 7: For fixed values of all other hyperparameters, the variation in EGWB concerning different initializations of the transport plan is shown. It's observed that including the joint distribution in the initializations leads to better results. This improvement might be attributed to the fact that the results of GWL and SpecGWL are close to certain local minima, and including the joint distribution allows exploration of more directions.

(Here, we employed $D = Heat[t]$, with $w = 0.5$ as a hyperparameter; similar conclusions arise with alternative settings.) Therefore, our subsequent experiments will focus exclusively on $\rho = 0.5$ or $1$.

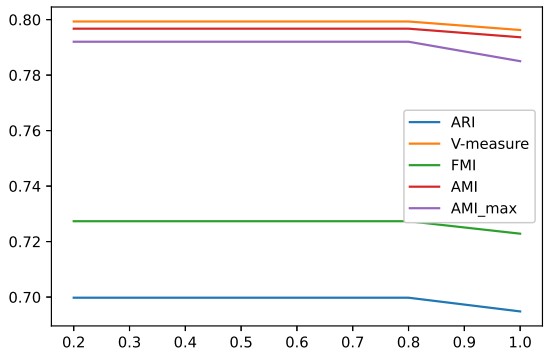

Figure 8: The performance of EGWB with different initializations of the transport plan varies, with better results observed when including the joint distribution. Additionally, it's noticeable that similar performance is achieved when $\rho$ is within the range of (0,1). Therefore, we can default to using $\rho = 0.5$ as it yields consistent results.

### B.3.2    SOURCE COST MATRIX $D$

When we directly use the adjacency matrix as the data graph's structural matrix in EGWB, the overall trend aligns with using the heat kernel. However, the overall performance is not as good as when using the heat kernel, as shown in Figure 9, which is consistent with the findings of SpecGWL.

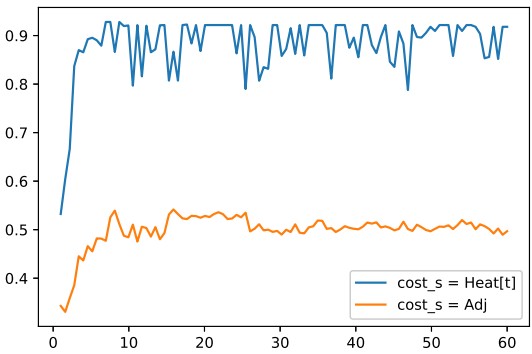

Figure 9: For using $\pi_{06}$ as the initial transport plan and considering both the adjacency matrix and the heat kernel as the graph's structural matrix, EGWB's ARI comparison indicates that the heat kernel performs better than the adjacency matrix.

### B.3.3    TARGET COST MATRIX $C_t$

If we fix $C_t$ to be the identity matrix, then it essentially becomes srGW, as it also relaxes the constraints on the target distribution $\mu$. This relaxation tends to yield better results compared to SpecGWL. However, srGW effectively introduces an exclusive lasso term, leading to clustering results where each class has an equal number of data points. Therefore, its performance is not as good as EGWB, as shown in Figure 10

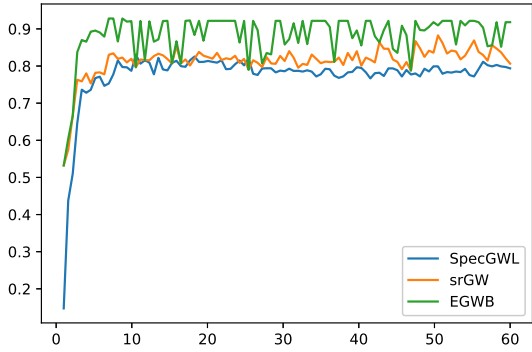

Figure 10: Comparison of AMI results between SpecGWL, srGW, and EGWB indicates that EGWB performs the best, while SpecGWL performs the worst. This aligns with our theoretical analysis, where SpecGWL and srGW are degenerate models of SpecGWL.

### B.3.4 PROPOSED HYPERPARAMETER $\lambda$

When $\lambda$ is exceedingly large, the concave regularization term dominates the objective function, rendering $f(\pi)$ a concave function. Consequently, Algorithm 1 rapidly converges to the extremal points, but this behavior heavily depends on the initial conditions. This is not the desired outcome. To address this issue, we employ a commonly used mathematical technique known as a continuation scheme, $\lambda = min(\frac{out_t{}^{\lambda_b}}{\lambda_a}, 1) * (\lambda + x)$, where $x$ is a very small number to make sure the greater condition in Theorem 2 (can be set equal to 0 in experiments). In this scheme, we gradually increase the value of $\lambda$ from a small one to a large value that satisfies convergence conditions. Intuitively, in the initial stages of the algorithm when $\lambda$ is small (close to 0), it is akin to employing classical entropy regularization methods, searching for more directions within the feasible set. As $\lambda$ increases, our algorithm eventually converges to an extremal point.

Inspired by enhanced balanced Min-Cut, we set $\lambda_a$ in the range of 10 to 20 and $\lambda_b$ in the range of 1 to 5. Figure 11 illustrates several possible combinations and the corresponding EGWB model's variation with respect to t. It appears that $(\lambda_a, \lambda_b) = (20, 5)$ is a particularly promising combination.

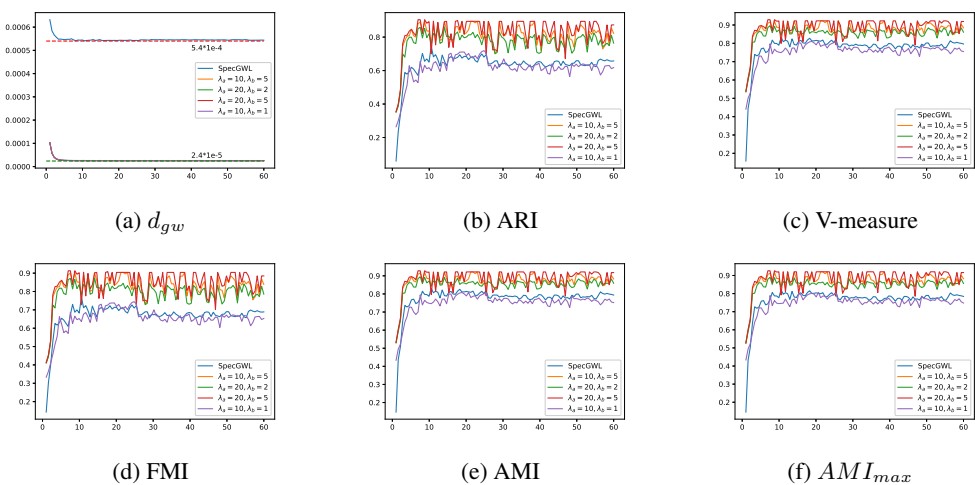

Figure 11: The different settings of $(\lambda_a, \lambda_b)$ effects on the EGWB.

### B.3.5 PRIOR NODE DISTRIBUTION $p_s$

Based on research into power-law transformations of data point distributions in GWL, we provide the following settings: $\mu_s = \frac{d_s}{\sum d_s}$, where $d_s = (deg(s) + a)^b$, we set $a = 0$ by default and $b \in [0, 1]$. Among these settings, when b equals 0, it corresponds to a uniform distribution, and when b equals 1, it is equivalent to using the degree to represent the node sizes. Figures 12 and 13 show the choice of b has little to no impact on our algorithm, which further validates our mass splitting technique.

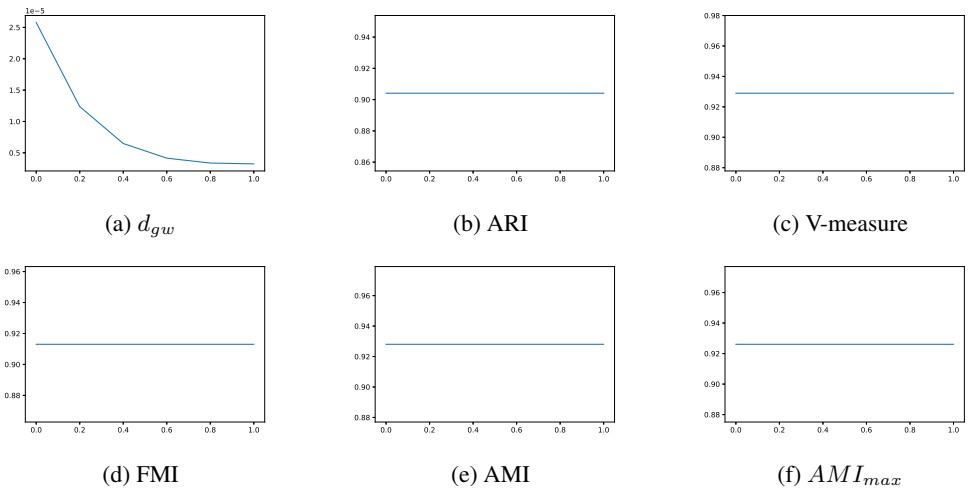

Figure 12: The different settings of $b$ effects on the EGWB in Indian dataset. when b equals 0, it corresponds to a uniform distribution, and when b equals 1, it is equivalent to using the degree to represent the node sizes. Using power law transformations can reduce the GW distance.

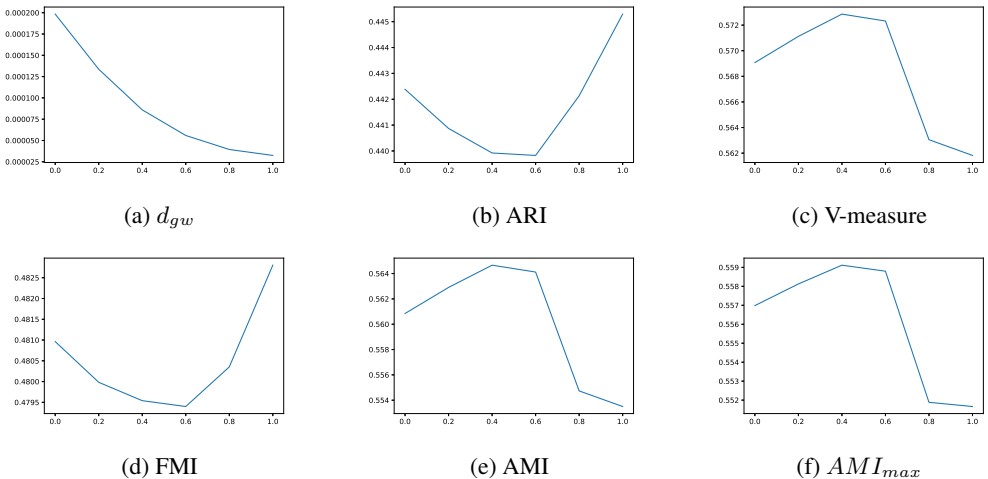

Figure 13: The different settings of $b$ effects on the EGWB in Wikipedia dataset. when b equals 0, it corresponds to a uniform distribution, and when b equals 1, it is equivalent to using the degree to represent the node sizes. Using power law transformations can reduce the GW distance.

