# OpenReview forum: "An Enhanced Gromov-Wasserstein Barycenter Method for Graph-based Clustering"
_ICLR.cc/2024/Conference — Submitted to ICLR 2024_

### Official Review · Reviewer_EjY1 · 2023-10-23

**Soundness:** 2 fair
**Presentation:** 2 fair
**Contribution:** 2 fair
**Rating:** 3
**Confidence:** 5

**Summary:**

Over a single input graph (D, h), authors study the problem of learning via the Gromov-Wasserstein loss (GW) a non-negative diagonal target structure D’ and its masses h’, in order to perform a partitioning of (D, h) via the underlying estimated (GW) transport plan. They propose to use a Block-Coordinate Descent algorithm that alternates between i) estimating a semi-relaxed GW transport plan using a mirror-descent scheme with an additional strictly concave regularization; ii) updating the diagonal structure D’ in closed-form. Authors empirically study the concavity of the resulting problem, and provide proofs of convergence for their algorithm. Then, they connect this GW (diagonal barycenter) problem or simplified variants to well-known clustering methods such as Min-Cut based methods, NMF and Max-Dicut. Finally, they study the relevance of their approach for graph partitioning of synthetic and real-world datasets.

**Strengths:**

-	Introduce a novel GW-based transport problem to perform graph partitioning.
-	Provide first results on the concavity of this problem. Then provide an algorithm to and introduce a strictly concave regularization of this problem that might help for such tasks.
-	Proof of convergence of their BCD algorithm.
-	Interesting but simple connections with existing clustering methods such as Min-Cut based methods , NMF and Max-Dicut.
-	Benchmark on GW-based and SOTA approaches for graph partitioning.

**Weaknesses:**

**Overall appreciation**: This paper tends to omit very similar recent works and the theoretical results seem either incomplete (analysis of concavity and relationships with existing graph partitioning/clustering methods) or incremental.

*NB: I added comments after authors' rebuttal in italic for each point*

- **1. Authors omit several recent contributions on GW**:
     - **a)** [A] studies the complete (sr)GW barycenter problem where the target structure is not forced to be a non-negative diagonal matrix. This paper shows that it is a SOTA method for spectrum-preservation graph coarsening and provide strong theoretical contributions supporting its use.
       **i)** These spectrum properties are also of particular interest for graph partitioning, hence the (sr)GW barycenter problem should be rigorously compared theoretically and empirically to the (sr)GW diagonal barycenter problem. **ii)** I believe that simple Stochastic Block Models (SBM) could provide a stress test over which the srGW diagonal barycenter problem fails contrary to the srGW barycenter problem : e.g using non-assortative SBM with a unique intra-cluster connectivity p smaller than inter-cluster connectivities $q_{cluster_i, cluster_j}$. Moreover I expect the diagonal structure to be more sensitive to contrasts within any SBM, i.e small variations between intra/inter-connectivities. Could authors perform such empirical sanity-checks ?

        [*None of these points have been clearly addressed by the authors during rebuttal*]

     - **b)** relations to srGW [Vincent-Cuaz et al, 2022] : **i)** The (sr)GW barycenter problem over a single input graph is a particular case of their dictionary learning. **ii)** The srGW solver proposed by authors is exactly the mirror-descent algorithm introduced in this other paper over which a concave regularization is added. These two points should be clearly stated in the paper.

        [*None of these points have been clearly addressed by the authors during rebuttal*]

     - **c)** On the proof of convergence for the algorithm: **i)**  [B, C] provide a scheme of proof to establish a non-asymptotic convergence of the regularized srGW solver. [*Not considered by the authors during rebuttal*]

         **ii)** An overview of the proof strategy for Theorem 3 should be clearly state  [*Not considered by the authors during rebuttal*]. More importantly Lemma 2 should be clarified : as such it seems wrong/ incomplete to me, e.g differentiability issues at the border are avoided, limits are considered out of the domain, continuity arguments are used without defining any topology etc...

         [*I am sorry I made a mistake on this matter, Lemma 2 is correct. As first suggested, an overview of the proof strategy would have helped for readability. Moreover authors follow a proof scheme from another more generic paper over which it would have been relevant to discuss relations. From my understanding, the first convergence proof for the srGW problem provides bounds involved in their finale convergence analysis. A sharper convergence proof - following B, C - could provide a sharper analysis and adaptive scheme for their regularization parameter.*]

         **iii)** The overall learning algorithm seems to be a particular case of two-block BCD well-studied in [D]. [*Reference not considered by authors. It implies that more generic converge proofs already exist for their BCD.*]

     - **d)** First parts of the supplementary materials: (minor) paragraph  ‘Non-convexity of GW discrepancy’ exposes known relations. (more important) paragraph ‘Assumption of uniform distribution’ seems to be a bad justification for the choice of input distributions that only translates the notion of weak-isomorphism discussed in [Chowdhury et al, 2019].

        [*Rebuttal made by authors is not compelling.  My point is that this dilution of mass// duplication of points of the support is absolutely not a justification for assuming a uniform distribution. What matter is the total mass assigned to the original point of the support. Your justification is misleading and formally wrong if you rigorously acknowledge the support of the measure. You can say we pick uniform distributions because it is the most common choice, note that there are other options (degrees etc..)... Moreover the sensitivity analysis in the supplemetary material is really not clear, that would be better to see a complete benchmark with same hyper parameter validation with several cases as b = 0 / b =1 / b in ]0, 1[. Maybe that is a by-product of learning the diagonal/complete barycenter, or just of diverse regularization coefficients. "which further validates our mass splitting technique." just does not make sense.*]


- **2. The several concavity analysis done by authors are incomplete and not conclusive:**
     - **a)** Could you detail the experiments illustrated in Figure 1 ? What is D’ in this setting ? What Is the initial transport plan used for these experiments ? Are these findings consistent w.r.t these initial transport plans (should be validated using the MCMC sampler in SpecGWL) ? What are the solvers used for these experiments ? If entropically regularized ones, please compare results to exact solvers such as conditional gradient solvers. *[Partially addressed by authors]*
    - **b)** Are these findings specific to heat kernels or do they generalize to PSD matrices e.g Laplacians  ? *[Misunderstood by authors - no time for discussions]*
    - **c)** None of the theoretical studies on the concavity of the overall learning problem are complete or convincing: **i)** proof/paragraph: "One common condition for extremal points" only shows that for an optimal target masses nu*, the resulting GW problem is concave hence solutions are extremities of admissible coupling with marginals mu and nu*. It does not show that extremities of the set of semi-relaxed couplings with first marginal mu, i.e hard-clustering matrices, are solutions. *[Incomplete analysis by authors maybe we misunderstood each other on the term extremities, considering that they always assume the existence of corners for $U(mu, nu^\star)$ no matter $nu^\star$ .]*

      **ii)** I do not see when the other condition in equation 19 could be applied, authors should discuss this point. *[partially addressed but still emphasises the strong specificity of this result]*.

      **iii)** Remark: Overall, a too recent contribution [E] to be taken into account at the submission date deals with these concavity problems for srGW barycenters and could guide authors to derive an analog result for the srGW diagonal barycenters.


- **3. Zero masses**: Authors do no mention the flexibility of this learning problem to get optimal target masses which are equal to zero and might allow to detect true number of clusters in some settings, as discussed in [Vincent-Cuaz et al, 2022]. *[partially addressed by authors]*


- **4. Missing points in experiments**:
    - **i)** Please benchmark methods in terms of running times too. Trade-off in terms of performances and speed should be explicit. *[partially addressed by authors]*
    - **ii)** The strongly concave regularization with continuation scheme proposed by authors introduce several hyperparameters. Please conduct an ablation study over this regularization. Plus could you complete Figure 11  that shows that the method is quite sensitive to these hyperparameters, with other datasets ? *[Ablation study not considered by authors. Sensitivity analysis apparently completed but from the revised paper version we can not even know what is the dataset used in these experiments]*
    - **iii)** Authors rely on other GW-based methods to get initial transport plans for their method. Whereas [Vincent-Cuaz et al, 2022] proposed to leverage k-means algorithm, which is a quite common technique in the clustering or graph partitioning literature. I guess, in concave setting solvers can be stuck at extremities, hence it would be relevant to force initial within the polytope e.g with kmeans + mu.nu^T. Could you further compare these choices ? *[not considered by authors]*

- **5. Some parts in Section 3.3 are not clear** and should be clarified: **i)** srGW to Identity: ‘This results in each cluster containing an equal number of data points.’ It clearly does not seem to be the case. **ii)** relation to NMF: does it really coincide with the srGW diagonal barycenter problem or rather with the complete barycenter problem ? *[addressed by authors]*




[A] Chen, Yifan, et al. "A Gromov--Wasserstein Geometric View of Spectrum-Preserving Graph Coarsening." International Conference on Machine Learning. 2023.

[B] Scetbon, M., Cuturi, M., & Peyré, G. (2021, July). Low-rank Sinkhorn factorization. In International Conference on Machine Learning (pp. 9344-9354). PMLR.

[C] Scetbon, M., Peyré, G., & Cuturi, M. (2022, June). Linear-time gromov wasserstein distances using low rank couplings and costs. In International Conference on Machine Learning (pp. 19347-19365). PMLR.

[D] Grippo, L., & Sciandrone, M. (2000). On the convergence of the block nonlinear Gauss–Seidel method under convex constraints. Operations research letters, 26(3), 127-136.

[E] Van Assel, Hugues, et al. "Interpolating between Clustering and Dimensionality Reduction with Gromov-Wasserstein." arXiv preprint arXiv:2310.03398 (2023).

**Questions:**

I invite the authors to discuss the above-mentioned weaknesses and to answer the questions (potentially implying additional experiments) I have associated with them in order to complete my development.

---

> ### Author Response · Authors · 2023-11-23
>
> 1.1 We sincerely apologize for overlooking this article. We appreciate the reviewer's excellent suggestions and valuable references. We will  conduct a more systematic study comparing these various methods. We also welcome the reviewer to engage in more detailed discussions and future study with us.
>  Regarding srGW, it can be considered a special case of EGWB where $D^{'}$ is fixed as the identity matrix. Additionally, the concave regularization introduced in EGWB is aimed at reducing $\| \pi \|_{L_0}$, increasing its sparsity for obtaining hard clustering results. In srGW, the concave regularization is intended to make $\nu$ sparser, reducing redundant clusters. Their designs have different intentions, leading to naturally different outcomes.
>
> 1.2 The proof of Lemma 2 is derived from reference [a]. If you harbor any inquiries, you may peruse the details. All functions in our paper exhibit continuous differentiability.
>
> 1.3 Kindly note that our intention was merely to elucidate that any GW problem can be reformulated under the assumption of $\mu$ follows the uniform distribution , and this does not imply that our algorithm is exclusively effective for uniform distributions. Our interest in this scenario stems from its simplicity and convenient explication of connections with other clustering methods, such as Min-Cut. The theorems we have proven can be effortlessly extended to encompass any distribution. Furthermore, as demonstrated in our experiments detailed in Appendix B 3.5, the utilization of power-law transformations of data point distributions as $\mu$ does not impact the final results of EGWB.
>
> 2.1 Figure 1 depicts the code using SpecGWL directly with the WIKI dataset, where all hyperparameters and initial values precisely match those of SpecGWL. The intention is to demonstrate that, after thresholding (i.e., selecting the index with the maximum element in each row as the clustering label), the GW distance is further reduced. Therefore, our EGWB focuses on directly seeking hard clustering.
>
> 2.2 All convergence proofs are applicable to all positive semi-definite matrices.
>
> 2.3 (a) We believe the reviewer has misunderstood our intent. We demonstrated that in the Monge GW problem (which is essentially finding the optimal mapping, corresponding to hard clustering), the setup is such that any $ \nu $ must satisfy the condition that each node can only belong to one class. Therefore, its Kantorovich relaxation problem is about minimizing a concave function over a convex set, where the solution naturally belongs to the extremal points of $ \Pi(\mu, \nu) $, it is also a part of the extremal points of $\Pi(\mu, \cdot)$. Hence, our problem setup involves using the extremal points of $ \Pi(\mu, \cdot) $ as the feasible set.
>         (b) If $ D $ and $ D^{'} $ are similarity matrices with values ranging between 0 and 1, it is natural that the diagonal of $ D $ is all 1's, satisfying this condition. However, ensuring that $D$ is a positive semi-definite matrix is an open question.
>
>
> 3  In theoretical analysis, by relaxing the weight constraints, the optimal value $ \nu $ can be sparse, implying that some classes may be empty sets. In such cases, we can set the corresponding $ D^{'}_{tt} $ to $-c$ in our paper, where $c$ is a large number (or 1 as discussed in srGW), ensuring that no nodes will be clustered in the $t$-th set in the next iteration. This is because
>
> $\sum_{i, i^{'}} \left(D_{ii^{'}}-D_{tt}^{'}\right)^{2} \pi_{i t} \pi_{i^{'} t}$ will be larger than any other component in $ \mathcal{E}_{D, D^{'}}(\pi) $.
>
> 4.1 The runtime is not orders of magnitude different because we employ a two-layer optimization, simultaneously optimizing both the target matrix and the transport plan. Each sub-optimization problem for the transport plan has a complexity almost identical to that of GWL or SpecGWL. However, our advantage lies in introducing higher degrees of freedom, theoretically allowing for better results in more complex scenarios.
>
> 4.2 We have presented a parameter ablation analysis in Appendices B 3.4, encompassing indices such as GW distance, AMI, ARI, V-measure, and FMI.
>
> 5.1 The objective function of srGW is equivalent to a standard Min-Cut function augmented by an exclusive lasso regularization term with a coefficient of $\frac{1}{2}$. The minimum value of this regularization term is achieved when the sizes of all classes are equal. You can find the proof in our Appendix B.1  for details.
>
> 5.2 According to our proof in Appendix B.1, page 21, for any square matrix, Weighted Symmetric NMF is equivalent to the complete Monge-type GW barycenter problem, which means the feasible set is constrained to be mappings represented by rectangular permutation matrices.
>
> [a] Razaviyayn, Meisam, Mingyi Hong, and Zhi-Quan Luo. "A unified convergence analysis of block successive minimization methods for nonsmooth optimization." SIAM Journal on Optimization 23.2 (2013): 1126-1153.

---

### Official Review · Reviewer_Zq9r · 2023-10-28

**Soundness:** 3 good
**Presentation:** 2 fair
**Contribution:** 2 fair
**Rating:** 6
**Confidence:** 3

**Summary:**

The paper propose a graph partitioning approach based on the Gromov-Wasserstein (GW) distance. This approach minimizes the GW distance between the target graph and an empty graph with fewer number of nodes. The optimization problem finds an optimal mapping, which determines the node clusters. It jointly optimizes node mass distribution and pairwise distance relations in the empty graph, the later constrained to be diagonal. The authors propose an algorithm which alternates the minimization of these variables and provide theoretical convergence guarantees.

Furthermore, the authors establish connections between their approach and existing GW-based methods, as well as alternative techniques like Min-Cut. Finally, they empirically demonstrate the effectiveness of their proposed method.

**Strengths:**

The authors present an extension of previous GW-based clustering methods, while connecting them with other approaches such as the Min-Cut or Non-negative Matrix Factorization. In addition, they propose an algorithm with theoretical guarantees. In this regard, the paper seems to be theoretically well founded.

Additionally, their proposed algorithm showcases remarkable robustness against edge noise when compared to the competing methods outlined in the paper.

**Weaknesses:**

- Challenges in Readability: In certain instances, the meaning of the notation, although not formally introduced, can be grasped from the context (e.g., $mathbb{I}_K$ denoting the identity matrix with $K$ rows). However, there are situations where the notation becomes confusing, posing a challenge to the paper's readability. For example, in the discussion of the "Monge's type Gromov-Wasserstein barycenter" in Section 3, the optimal mapping matrix is denoted as $MGW(G, G′)$, but this notation is also used as the objective in the minimization equation (7). The concept of minimizing a matrix raises confusion. In addition, the symbols $\pi$ and $\Gamma$ are interchangeably used to refer to the same object. For instance, three lines before equation (7), it states $\nu=\pi^T 1_N$, but in this context, as far as I did not misunderstand it, we are assuming a hard clustering mapping and therefore $\Gamma$ should be the appropriate symbol. Furthermore, though it is not crucial, adding the labels to the x-axis and y-axis of the plots would also ease the readability of the figures.

- The proposed method initialization depends on the results of other methods such as GWL and SpecGWL.

**Questions:**

- Initialization dependency: The paper mentions using a linear combination of GWL, SpecGWL, and joint distribution results as initial values for EGWB. However, it's unclear how the algorithm relies on this initialization. Could a less informed start, like the uniform distribution, yield comparable results or does one need to start from a relatively good initialization?
- Computational cost: While the paper outlines the computational cost per iteration, the average number of iterations required for convergence remains undisclosed. Additionally, considering the initialization dependency, it's crucial to know the overall time needed to run EGWB, especially if solutions for GWL and SpecGWL must be computed beforehand.
- Synthetic data: Figure 4 exhibits superior results for GWL and SpecGWL compared to Figure 2. Moreover, in Figure 4 GWL out performs SpecGWL given the true cluster size distribution. Is there any reason why is that the case? I am actually surprised that, for these apparently simple problems GWL and SpecGWL fail to retrieve the right clustering. Understanding the specific reasons for their failure is beneficial in order to comprehend why EGWB, in contrast, succeeds.
- Number of clusters: A parameter that needs to be set is the expected number of clusters $K$. This is indicated by the number of nodes in the empty graph. How robust is the algorithm to the choice of $K$? Particularly intriguing is the scenario where $K$ exceeds the actual number of clusters. In theory, it is possible that the optimal mapping does not assign any mass to the extra nodes of the empty graph. In that case, the algorithm would still be able to retrieve the true partition. Does this happen in practice, and how does the algorithm adapt to such situations?

---

> ### Author Response · Authors · 2023-11-23
>
> We appreciate your insightful observations and constructive criticism.
> 1. Thank the reviewer for point out those typos, and in response, these have been incorporated in the revised paper. Specifically, (i) MGW represents the Monge-type GW distance induced by the optimal mapping $\Gamma$. It also depends on the target graph $G^{'}$, which is composed of a matrix $D^{'}$ and node distribution $\nu$. Our objective is to find the optimal graph $G^{'}(D^{'},\nu)$. (ii) Figure 3  illustrates the AMI results of SpecGWL, srGW, and EGWB across various heat kernel times (t) for the WIKI dataset.  This figure serves as an illustrative representation of our motivation for delving into graph-based hard clustering.
>
> 2. In fact, we thoroughly elaborate on this experiment in the appendix. It is crucial to recognize that this challenge poses a non-convex optimization problem. Consequently, selecting a  well-chosen initial value undoubtedly improves the optimality of the final convergence point to a certain extent.
>     Both GWL and SpecGWL provide a transport plan $ \pi $ in the joint distribution space. Using the combinations of these plans as initial points is a natural idea. Moreover, if a uniform distribution is directly employed as the initial value, theoretically, it is equivalent to SpecGWL in the early stages of the algorithm.
>
> 3. The theoretical prediction of average numbers of iterations is still open for such non-convex optimization problems. However, in experiments, we observe that during the inner optimization process for $\pi$, approximately three hundred iterations are required to reduce the relative error, denoted as $ \|\frac{\pi^{t+1} - \pi^{t}}{\pi^{t}}\|_F $, from $ O(1)  $ to $ O(1 \times 10^{-4}) $.
>
> 4. Indeed, in our experiments, SpecGWL achieved a smaller GW distance compared to GWL. The crucial point is that both methods employ a fixed $G^{'}(D',\nu)$ and then learn the transport map, which makes the clustering results less convincing. In contrast, our EGWB treats it as a variable for optimization, introducing more degrees of freedom that are determined by the data, naturally yielding superior results.
>
> 5. In the graph-based clustering experiments conducted, $ K $ is a pre-determined parameter. However, in theoretical analysis, by relaxing the weight constraints, the optimal value $ \nu $can be sparse, implying that some classes may be empty sets. In such cases, we can set the corresponding $ D^{'} $ to a relatively large constant (eg, 1), as discussed in reference [a]. In our numerical experiments, we treat $K$ as a prescribed parameter such that all methods are compared on the fair stage.
>
>
>
> [a] Cédric Vincent-Cuaz, Rémi Flamary, Marco Corneli, Titouan Vayer, and Nicolas Courty. Semi-relaxed gromov-wasserstein divergence and applications on graphs. In International Conference on Learning Representations, 2022.

---

### Official Review · Reviewer_iAAc · 2023-10-31

**Soundness:** 2 fair
**Presentation:** 2 fair
**Contribution:** 2 fair
**Rating:** 3
**Confidence:** 3

**Summary:**

In this submission, the authors propose a new graph partitioning framework EGWB, which relaxes the target structure and distribution constraints in Gromov-Wasserstein Learning (GWL) with a class of positive semi-definite matrices.
In particular, by learning the target structure matrix associated with the transport plan, the authors extend the GWL framework to a special GW barycenter problem (with only one graph), which enhances the flexibility of the GWL framework.
The proposed method is shown to be effective according to empirical results in various graph partitioning tasks.

**Strengths:**

Graph partitioning based on utilization of the Gromov-Wasserstein (GW) distance is an interesting and significant problem.

**Weaknesses:**

1. How to initialize $D’(0)$? How to set the value of $K$?

2. It seems the authors confuse the task of graph partitioning and that of graph clustering. They muddle up partitioning and clustering throughout this paper.
I think the experiments in section 4.2 are more likely to be a graph partitioning task, rather than graph clustering, as is claimed by the authors. Please use one of the two definitions consistently in the paper.

3. In the subsection of Results and Discussion, the authors say they employ five metrics, however, I only find AMI. If they take the results reported in appendix into account, then should clarify this in the main context.

4. Are the datasets in section 4.2 asymmetric or symmetric? Do the authors symmetrize the directed graphs?

5. What do the two axes in Fig.3 represent? It should be labeled in the figure.

6. There are typos and careless statements and the authors need to polish this paper carefully. For example, 1) page 1, it should be $G_1(D_1,P_1)$ in the third row from the bottom, 2) page 2, the second paragraph, the second point of the limitations has grammatical mistakes, 3) What is EGWB an abbreviation for? The author put forward EGWB without any explanations in page 6.

**Questions:**

Please see above.

---

> ### Author Response · Authors · 2023-11-23
>
> We appreciate some of the review's comments while cannot agree with the others. We do not think the reviewer carefully read and correctly understand our paper. Thus his/her comment is subjective and not fair.
>
> 1. The  $D^{'}(0)$ can be readily derived from equation (10) of our paper, provided one thoroughly peruses our document. Anyway, we can explain it more in detail here:
>     When we have an initial guess $\pi(0)$, we employ $$\frac{\pi^{\top}_{:,j}(0)D\pi_{:,j}(0)}{\pi^{\top}_{:,j}(0)\pi_{:,j}(0)}$$ as the diagonal of  $D^{'}(0)$.  The selection of $K$ follows standard procedures as outlined in any clustering textbook. Consequently, in our paper, we treat $K$ as a predetermined parameter, ensuring a fair comparison across all methods.
>
> 2. The main focus of our paper is the clustering based on graph structure. This is clearly stated in section 3.1, where our aim is to solve the hard clustering problems via graph partitioning. The point of doing clustering via graph partitioning has already been made in the literature prior to our paper, see [a][b].
>
> 3. Once more, it appears that the reviewer has not meticulously perused our paper. We have meticulously presented comprehensive numerical results in the appendices, encompassing indices such as ARI, V-measure, and FMI. Nevertheless, to further address this concern, we will augment the main text with a clarifying sentence.
>
> 4. The datasets in Section 4.2 consist of undirected graphs, rendering their adjacency matrices naturally symmetric. In instances involving directed graphs, we opt for the normalized Laplacian, elucidated within the framework of random walks [c]. In all of our numerical experiments, we use the natural adjacency matrices without any artificial symmetrization in order to make our comparison fair. This choice is deliberately articulated in the supplementary materials appended in the appendices.
>
> 5. Figure 3  illustrates the AMI results of SpecGWL, srGW, and EGWB across various heat kernel times (t) for the WIKI dataset.  This figure serves as an illustrative representation of our motivation for delving into graph-based hard clustering.   We still thank the reviewer for pointing out this ambiguity, and in response, corrected labels have been incorporated in the revised paper.
>
> Thank the reviewer for point out those typos. They have been corrected.
>
>
> [a] Hongteng Xu, Dixin Luo, and Lawrence Carin. Scalable gromov-wasserstein learning for graph
> partitioning and matching. Advances in Neural Information Processing Systems, 32, 2019a.
>
> [b] Samir Chowdhury and Tom Needham. Generalized spectral clustering via gromov-wasserstein
> learning. In International Conference on Artificial Intelligence and Statistics, pp. 712–720. PMLR,
> 2021.
>
> [c] Fan Chung. Laplacians and the Cheeger inequality for directed graphs. Annals of Combinatorics, 9(1):1–19, 2005.

---

### Official Review · Reviewer_2sWH · 2023-11-01

**Soundness:** 4 excellent
**Presentation:** 3 good
**Contribution:** 3 good
**Rating:** 6
**Confidence:** 3

**Summary:**

The paper presents a Gromov Wasserstein (GW) Clustering Method based on a single marginal GW Barycenter : EGWB. The method also allows to add marginal or ambient metric constraints on the barycenter. Thus effectively showing that their method is a generalization of existing GW Learning methods. Later on more precise links are made with existing methods.
They first introduce a Monge type barycenter problem and a Kantorovich relaxation of it. It is shown that under appropriate conditions the two problems are equivalent.
An optimization algorithm relying on entropic regularization is presented. The convergence of the algorithm is shown.
Finally their method is benchmarked against existing GW Clustering methods, on synthetic and real data. On all accounts EGWB outperforms existing GW methods.

The contributions are the following:
- Introduced a generalization of existing GW Learning Methods
- Demonstrated theoretically and empirically their algorithm for solving the problem converges and has state of the art performances

**Strengths:**

The paper presents a unifying framework for GW methods. It does so clearly.

The core idea is to introduce the GW barycenter problem and note that adding constraints recover existing methods. This problem seems novel in that context and they address with clarity the first questions one can have  : equivalence between the Monge and Kantorovich type formulations, link with other methods in GW Learning as well as in Graph partitioning.

Their algorithm is an alternating minimization one. However they address the non convexity of the transport plan update by using an interesting combination of existing regularization methods : entropic regularization, link with the Wasserstein Barycenter problem which has better structure.

The synthetic data example is informative of how the barycentric nature of the problem allows for more efficient clustering. In the analysis of the performance on real data the explanation of the performance in relationship with the structure of the data is appreciated.

**Weaknesses:**

In the paragraph about Kantorovich relaxation it is stated that the minimum is attained at an extremal point under some conditions which are detailed in appendix. This point is central to the use of the algorithm afterwards. Thus I believe the conditions should be put forward in the main text.

In theorem 3 it is unclear in which case the algorithm converges with entropic regularization, however this is central to showing that the implemented algorithm does converge.

**Questions:**

How does the result of theorem 3 relates to the convergence of the algorithm implemented in practice?

Are there a stability result of the limit of the algorithm/solution of the problem with respect to the epsilon parameter?

In practice what are the optimal value used for the epsilon parameter for each datasets?

---

> ### Author Response · Authors · 2023-11-23
>
> We appreciate your insightful observations and constructive criticism.
> 1. 1st point in the weakness comment:
> We put forward the condition (19) in the last paragraph in section 3.1 of the main text while leaving the proof in the appendix.
> 2. Theorem 3 provides a conservative condition between $\lambda$ and $\epsilon$ under which the convergence of the alternating algorithm is guaranteed. In practice, we observe that even this condition is violated, the convergence still holds. Anyway, this condition roughly offer a general guideline on selecting practical parameters in order to make the algorithm work.
> 3. The stability result with respect to $\epsilon$ is still on demand. However, we numerically found that $\epsilon$ does affect the clustering results. This is commented in detail in the next item.
> 4. We found that the results were sensitive to the choice of regularization parameter $\epsilon$ (as is also observed by [a][b]), leading to numerical blowups if not chosen carefully. In reporting each of the results below, we hand-tuned $\epsilon$ as $\epsilon=2e-5$ for WIKI and INDIAN datasets, and $\epsilon=3e-6$ for EU-email and AMAZON datasets.
>
>
> [a] Hongteng Xu, Dixin Luo, and Lawrence Carin. Scalable gromov-wasserstein learning for graph
> partitioning and matching. Advances in Neural Information Processing Systems, 32, 2019a.
>
> [b] Samir Chowdhury and Tom Needham. Generalized spectral clustering via gromov-wasserstein
> learning. In International Conference on Artificial Intelligence and Statistics, pp. 712–720. PMLR,
> 2021.

---

### Meta-Review · Area_Chair_gc9S · 2023-12-06

**Metareview:**

Based on the reviews, it is clear that the paper has interesting aspects, such as the introduction of a novel Gromov-Wasserstein (GW)-type method for graphs, alongside theoretical analysis and extensive numerical evaluation. However, there are significant areas that require improvement. A major shortcoming identified by the reviewers is the lack of an in-depth discussion and comparison with existing works closely related to GW, particularly in the context of GW barycenters. Moreover, several aspects of the theoretical results are noted to require detailed rewriting to enhance clarity and comprehensibility. While the authors have addressed some of these issues in their rebuttal, the reviewers unanimously feel that the paper necessitates extensive rewriting. In light of these considerations, the paper, in its current form, does not seem ready for publication.

**Justification For Why Not Higher Score:**

GW is now well established, the paper does not includes enough citations/comparison to related works.

**Justification For Why Not Lower Score:**

N/A

---

### Decision · Program_Chairs · 2024-01-16

Reject